# Cytotoxicity and Bioactivity of Dental Pulp-Capping Agents towards Human Tooth-Pulp Cells: A Systematic Review of In-Vitro Studies and Meta-Analysis of Randomized and Controlled Clinical Trials

**DOI:** 10.3390/ma13122670

**Published:** 2020-06-12

**Authors:** Mariano S. Pedano, Xin Li, Kumiko Yoshihara, Kirsten Van Landuyt, Bart Van Meerbeek

**Affiliations:** 1Department of Oral Health Sciences, KU Leuven (University of Leuven), BIOMAT—Biomaterials Research Group & UZ Leuven, University Hospitals Leuven, 3000 Leuven, Belgium; simon.pedano@kuleuven.be (M.S.P.); xin.li@kuleuven.be (X.L.); kirsten.vanlanduyt@uzleuven.be (K.V.L.); 2National Institute of Advanced Industrial Science and Technology (AIST), Health Research Institute, 2217-14 Hayashi-Cho, Takamaysu, Kagawa 761-0395, Japan; k-yoshi@md.okayama-u.ac.jp; 3Department of Pathology & Experimental Medicine, Dentistry and Pharmaceutical Sciences, Graduate School of Medicine, Okayama University, 2-5-1 Shikata-cho, Kita-ku, Okayama 700-8558, Japan

**Keywords:** vital pulp therapy, biomaterials, calcium hydroxide, MTA, bioceramics, human dental pulp cells, hydraulic calcium-silicate cements

## Abstract

*Background.* In the era of biology-driven endodontics, vital pulp therapies are regaining popularity as a valid clinical option to postpone root-canal treatment. In this sense, many different materials are available in the market for pulp-capping purposes. *Objectives.* The main aim of this systematic review and meta-analysis was to examine literature regarding cytotoxicity and bioactivity of pulp-capping agents by exposure of human dental pulp cells of primary origin to these materials. A secondary objective was to evaluate the inflammatory reaction and reparative dentin-bridge formation induced by the different pulp-capping agents on human pulp tissue. *Data sources.* A literature search strategy was carried out on *PubMed*, *EMBASE* and the *Web of Science* databases. The last search was done on 1 May 2020. No filters or language restrictions were initially applied. Two researchers independently selected the studies and extracted the data. *Study selection included eligibility criteria, participants and interventions, study appraisal and synthesis methods.* In vitro studies were included when human dental pulp cells of primary origin were (in) directly exposed to pulp-capping agents. Parallel or split-mouth randomized or controlled clinical trials (RCT or CCT) were selected to investigate the effects of different pulp-capping agents on the inflammation and reparative bridge-formation capacity of human pulp tissue. Data were synthesized via *odds ratios* (95% confidence interval) with fixed or random effects models, depending on the homogeneity of the studies. The *relative risks* (95% confidence interval) were presented for the sake of interpretation. *Results.* In total, 26 in vitro and 30 in vivo studies were included in the systematic review and meta-analysis, respectively. The qualitative analysis of in vitro data suggested that resin-free hydraulic calcium-silicate cements promote cell viability and bioactivity towards human dental pulp cells better than resin-based calcium-silicate cements, glass ionomers and calcium-hydroxide cements. The meta-analysis of the in vivo studies indicated that calcium-hydroxide powder/saline promotes reparative bridge formation better than the popular commercial resin-free calcium-silicate cement Pro-Root MTA (Dentsply-Sirona), although the difference was borderline non-significant (*p* = 0.06), and better than calcium-hydroxide cements (*p* < 0.0001). Moreover, resin-free pulp-capping agents fostered the formation of a complete reparative bridge better than resin-based materials (*p* < 0.001). On the other hand, no difference was found among the different materials tested regarding the inflammatory effect provoked at human pulp tissue. *Conclusions.* Calcium-hydroxide (CH) powder and Pro-Root MTA (Dentsply-Sirona) have shown excellent biocompatibility in vitro and in vivo when tested on human cells and teeth. Their use after many years of research and clinical experience seems safe and proven for vital pulp therapy in healthy individuals, given that an aseptic environment (rubber dam isolation) is provided. Although in vitro evidence suggests that most modern hydraulic calcium-silicate cements promote bioactivity when exposed to human dental pulp cells, care should be taken when these new materials are clinically applied in patients, as small changes in their composition might have big consequences on their clinical efficacy. *Key findings (clinical significance).* Pure calcium-hydroxide powder/saline and the commercial resin-free hydraulic calcium-silicate cement Pro-Root MTA (Dentsply-Sirona) are the best options to provide a complete reparative bridge upon vital pulp therapy. *Systematic review registration number. PROSPERO registration number:* CRD42020164374.

## 1. Introduction

Dental pulp-capping agents are defined as those materials used as a protective layer to an exposed tooth pulp to allow the tissue to recover and maintain its normal function and vitality [1,2]. Ideally, those materials should not only be inert, in the sense that they should not be toxic to the pulp cells, but they should be “bioactive” towards the tissues by stimulating migration, proliferation and osteogenic differentiation of the cells [3,4].

The tenets of minimal-invasive dentistry have caused a paradigm shift in the treatment of deep caries and vital pulp therapies. In this way, from total caries-excavation techniques, we have moved onto partial caries-excavation to avoid pulp exposures [5,6,7,8]. Something similar is happening in the treatment of reversible and irreversible pulpitis. Increasing evidence is showing that in the presence of strict aseptic conditions (rubber dam isolation) and with the aid of magnification, partial or full pulpotomy can serve as valid and less invasive alternatives to root-canal treatment [8,9,10,11]. This might have many advantages, since root-canal treatment is a more technically demanding and time-consuming treatment than (partial) pulpotomy [12].

Since the introduction of Pro-Root MTA (MTA; Dentsply-Sirona, Konstanz, Germany), the first hydraulic calcium-silicate cement developed, many other materials with similar compositions have been introduced into the market [13,14,15]. The main reasons for the increase of marketed materials are the good results obtained with MTA in terms of biocompatibility and long-term survival [16,17], and the need for materials with improved handling properties, lesser discoloration risks, better sealing abilities and reduced prices [18,19]. The latest developments in this search for improved dental pulp-capping agents are the resin-based calcium-silicate cements [20,21,22]. These materials possess enhanced handling properties by setting on command; they reduce the risk of discoloration; and, by optimizing the monomer composition, they may also adhere to tooth structure, by which improved sealing capacity can be expected [23,24,25]. Moreover, by adding resins to their composition, we may also better adhere them to resin composites and resin-modified glass ionomers, being put on top; reduce the treatment time; and reduce the risks of leakage and early filling loss [26]. However, the main drawback of this type of cement is the lack of biocompatibility of the monomers in contact with vital pulp tissue, which may hamper the formation of a complete hard tissue barrier at the exposed area [27,28,29]. Nevertheless, resins are not toxic by definition and many researchers are already working to develop biocompatible, naturally derived resin blends that may be suitable for biomedical applications [30,31,32,33,34]. These new type of photocurable resins are only prototypes but they have already been tested in in vitro and in vivo studies with promising results [35,36,37].

The *Guide to Clinical Endodontics* from the *American Association of Endodontists* recommends pulpotomy in permanent teeth only as an emergency or interim procedure until further root-canal treatment can be accomplished. However, recent randomized clinical trials have challenged this concept, as pulpotomy might be a successful treatment option for teeth with symptomatic irreversible pulpitis, even in cases with periapical involvement [11,38,39,40,41,42]. As this type of treatment is becoming more scientifically supported, a recent review and position statement article from the *European Society for Endodontology* [8] opened the door for this paradigm shift and clinicians are starting to perform such treatments [43,44,45]. In this way, the best available evidence suggests that the materials of choice for vital pulp therapy are calcium-hydroxide or MTA [46]. However, these materials have many side effects. The main problem of calcium hydroxide is its high solubility, which will create a gap between pulp tissue and final restorative material [47]. For MTA, the main drawbacks are: (1) risk of discoloration, (2) long-setting time and (3) difficult handling [19]. Therefore, recently introduced materials are gaining popularity among clinicians; for example, tricalcium-silicate cements such as Biodentine (Septodont, Saint-Maur-des-Fossés, France), and resin-based calcium-silicate cements, such as Theracal LC (Bisco, Schaumburg, IL, USA) and Biocal Cap (Harvard, Hoppegarten, Germany). However, very limited information is available regarding their biocompatibility when they are exposed to human dental pulp cells and tissue.

Recently, many reviews have been written about dental pulp-capping therapies and materials [48,49,50,51]. However, concerningly, this is the first systematic review aiming to compare all kinds of pulp-capping agents, including resin-based materials. Moreover, by using an indirect meta-analytical approach, we have tried to shed some light on the controversy that some studies found no difference in terms of (long-term) survival and reparative bridge formation between calcium-hydroxide materials and calcium-silicate cements [46,52,53], while others showed that calcium-silicate cements improved the prognosis of vital-pulp therapies [49,50].

Therefore, the aim of this study was to assess the in vitro biocompatibility of dental pulp-capping materials for vital pulp therapy when exposed to human dental pulp cells. As a secondary objective, in vivo studies were reviewed for inflammatory reaction and the presence of reparative dentin formation after direct exposure of the pulp tissue of completely developed permanent teeth to pulp-capping materials.

The null-hypotheses tested were (1) that there is no difference in in vitro biocompatibility for the different pulp-capping agents when exposed to human dental pulp cells; (2) that there is no difference in the short-term (<30 days) inflammatory reaction caused by the materials tested in vivo; and (3) that there is no difference in complete hard-tissue bridge formation in vivo after 30 days among the different materials tested.

## 2. Material and Methods

### 2.1. Protocol and Registration

The methodology of this review was based on the PRISMA (Preferred Reporting Items for Systematic Reviews and Meta-Analyses) guidelines [54,55]. This review was registered at the PROSPERO database (number: CRD42020164374).

The protocol for this review was designed by the authors with the support of an expert librarian from the Biomedical Sciences group of KU Leuven.

### 2.2. Eligibility Criteria

The eligibility criteria were different depending on the type of study (i.e., in vitro vs. in vivo studies) and the specific characteristics (inclusion and exclusion criteria) for each type of study are shown in Table 1.

### 2.3. Information Sources

A literature search was performed using the *PubMed*, *EMBASE* and *Web of Science* databases (Appendix A). The first search was performed on 1 October 2019 and was updated for the last time on 1 May 2020. No filters were applied. After removing duplicates with Endnote X9 software (Clarivate Analytics), we chose studies starting from 1993, because it is the date when mineral trioxide aggregate was patented, and together with calcium hydroxide it is considered as gold-standard material for pulp-preserving procedures. Finally, a manual search was conducted from the reference lists of relevant review articles published in the last 5 years.

### 2.4. Search Strategy

The search strategy was designed by 2 reviewers (MSP and XL) in collaboration with an expert librarian. As the main purpose of the review was not to find a specific answer to a specific clinical question, we decided to conduct the literature search using the main terms of interest, instead of the classic PICO structure. The main terms of interest chosen were: (1) “*Biocompatibility or pulp-tissue reaction*”; (2) “*Pulp-capping materials or agents*”; and, (3) *“(Human) dental pulp cells/tissue or (human) teeth.”* The complete search strategy used in the *PubMed*, *EMBASE* and *Web of Science* databases can be found in the Appendix A.

### 2.5. Study Selection and Data Collection Process

Studies were selected and data collected by two independent investigators (MSP and XL) who revised the full list of articles and selected the papers that were potentially of interest, first by title and then by abstract screening. Later, texts were fully screened to identify the articles that met the inclusion criteria. In case of disagreement, differences were discussed until agreement was reached. Only articles published in the English language were chosen (Figure 1).

Data extraction was done separately for in vitro (Table 2) and in vivo studies (Table 3 and Table 4). For the in vitro studies, the following data were obtained from the selected articles: (1) study characteristics: authors and year of publication, (2) materials tested, (3) type of exposure (direct/indirect) and the use of fresh or set materials, (4) parameters tested, (5) methods used and (6) results obtained. For the in vivo studies, the data collected were the following: (1) study characteristics: authors and year of publication, (2) type of study: randomized controlled trial (RCT) or controlled clinical trial (CCT), (3) method used for hemostasia, (4) materials tested, (5) etching of pulp tissue, (6) evaluation time, (7) presence (or not) of an independent examiner of the histological samples, (8) characteristics of the bridge formed, (9) characteristics of the inflammatory reaction and (10) amount and type of teeth used and age of the patients. When the data from the articles were unclear or could not be found, we contacted the authors by e-mail or ResearchGate. If no answer was received, the articles were excluded.

### 2.6. Risk of Bias in Individual Studies

For the risk of bias of in vivo studies and for the meta-analysis, the *Cochrane Handbook for Systematic Reviews of Interventions 6* [109] and the *Review Manager 5.3* software [110] provided by the Cochrane collaboration (www.cochrane.org) were followed. The risk of bias of each individual study can be found next to each forest plot in the meta-analysis (Figure 2, Figure 3, Figure 4 and Figure 5).

### 2.7. Data Analysis

#### 2.7.1. Data Synthesis

For the in vivo studies, quantitative analysis was performed with Review Manager 5.3 (Revman) software provided by the Cochrane Collaboration (www.cochrane.org). Dichotomous data were presented in forest plots as odds ratios (OR) with 95% confidence intervals (CIs) (Figure 2, Figure 3, Figure 4 and Figure 5). When the heterogeneity of the studies was considered “*low*” (I^2^ < 50%), “*fixed effects*” were considered. When heterogeneity was “*high*” or when it was not possible to measure (no direct comparisons available), “*random effects*” were evaluated. The difference in the effect between different materials was considered statistically significant when *p* < 0.05. Instead of the “*odds ratio*,” the “*relative risk*” (RR) was presented in Table 5, Table 6, Table 7 and Table 8 for the sake of simplicity in the interpretation of the results [109].

#### 2.7.2. Heterogeneity Assessment

The chi-square test and I^2^ statistic were used to assess heterogeneity. The fixed-effect model is suitable to estimate the typical effect for studies with low heterogeneity (I^2^ < 50%), whereas the random-effects model is used to assess the average distribution for studies with substantial unexplained heterogeneity (I^2^ ≥ 50% or *p* ≤ 0.05) [109].

#### 2.7.3. Assessment of Publication Bias

If more than 10 articles were included, publication bias was analysed by visual inspection of funnel plots. An asymmetrical distribution of funnel plot data may suggest the possibility of publication bias [109].

#### 2.7.4. Summary Measures

For the parameter “*inflammation*” of the included in vivo studies, inflammation was noted only when it was scored as severe or when the pulp tissue was defined as necrotic or abscess formation (severe inflammation and/or necrosis/abscess formation were considered as “*event*” in the Revman 3.5 software). As the parameter “*inflammation*” is dynamic (it might change over time), it was evaluated separately for different time points (i.e., inflammation up to 7 days, at day 15 and at day 30). Periods longer than 30 days were not taken into account, as inflammation due to material toxicity occurs normally within a short-day range. Inflammation occurring at longer time periods may be due to bacterial infiltration or trauma.

To evaluate the parameter “*bridge formation*” of the included in vivo studies, only the presence of a complete bridge was taken into account. When the bridge was incomplete or not present, this was considered as an “*event*” in the Revman 3.5 software.

The results were expressed as the *odds ratios* and 95% CIs (the “*relative risk*” (RR) was presented in Table 5, Table 6, Table 7 and Table 8 for the sake of simplicity in the interpretation of the results). When possible, heterogeneity of the studies and bias publication were also recorded. As many materials have been tested over time and no direct comparison was always available between them, an indirect meta-analysis was performed. Moreover, a network meta-analysis graph was made for all the available studies and their respective interactions (Figure 6).

### 2.8. Statistical Analysis

We conducted a meta-analysis to obtain estimates of the relative effectiveness of all interventions on the primary outcome by combining direct and indirect evidence using a fixed-effects or random-effects model.

### 2.9. Risk of Bias Across Studies (Certainty in the Evidence)

The quality of the evidence was assessed using the “*Grading of Recommendations Assessment, Development and Evaluation*” (GRADE) approach at the outcome level for each comparison between interventions [111]. The certainty in the evidence can be high, moderate, low or very low. When the certainty is derived from direct comparisons, randomized controlled trials provide high-certainty evidence. However, some issuese such as serious risk of bias, imprecision, inconsistency or publication bias can reduce the certainty [111].

## 3. Results

### 3.1. Search Results and Study Selection

The PRISMA flowchart with the search results and the study selection process can be seen in Figure 1. The *PubMed*, *EMBASE* and *Web of Science* searchs provided 6490, 4369 and 5682 results, respectively. The total sum of 16,541 articles was stored in a reference manager (Endnote X9, Clarivate Analytics). Twelve results found by hand searching through the reference lists of the articles and from other sources were added. Duplicates were removed manually with Endnote X9 (Clarivate Anaytics) reference manager, resulting in 10,469 unique articles. From these 10,469 studies, 1536 were excluded because they were published before 1993 (date of the first publication on MTA [15]). From the remaining 8933 articles, 7628 and 1117 articles were excluded as being non-relevant by screening the titles and abstracts, respectively. Finally, 188 articles were eligible for full-text screening (Figure 1). The year of publication ranged from 1946 to 2020 (later reduced to 1993–2020). The selection process is detailed in the PRISMA flow chart (Figure 1). Two in vivo studies that met de inclusion criteria were finally not included (after contacting the authors) in the qualitative/quantitative analysis because the data had been partially used in previous studies (Figure 1) [112,113]. Finally, 56 articles were included in the analysis, 26 involving in vitro studies and 30 in vivo studies. The reasons for exclusion are listed in Figure 1 and a full list of excluded articles (with reason for exclusion) is provided in Appendix A. Moreover, these 30 in vivo studies were included for quantitative meta-analysis, among which there were 10 controlled clinical trials (CCT) and 20 randomized controlled trials (RCT). The sample sizes of these studies were n = 355 (premolars) for immature permanent teeth and n = 657 for mature permanent teeth (301 molars and 356 premolars) for a total of 1012 teeth in all 30 included in vivo studies.

### 3.2. Study Characteristics

The characteristics of the included in vitro and in vivo studies are listed in Table 2, Table 3 and Table 4. The in vivo studies were subdivided depending on whether they included mature or immature permanent teeth (Table 3 and Table 4, respectively).

#### Risk of Bias within Studies

The risk of bias of the in vivo studies is presented along with the meta-analysis (forest plots; Figure 2, Figure 3, Figure 4 and Figure 5) and was determined following the Cochrane recommendations using the software Revman 5.3 (Cochrane). In general, except for two studies [85,100], most of the included in vivo studies were deemed as having high risk of bias. The most frequent types of bias were *selection and performance biases*, as in most studies no “*random sequence generation*” nor “*allocation concealment*” were described. It is true that “*blinding of participants and personnel*” is difficult to achieve as the materials have different appearances. Therefore, when this was the only risk of bias present, we did not consider it a high bias risk. However, this was the case in only two out of 30 studies.

### 3.3. Synthesis of Results

The results for each individual in vitro and in vivo study are presented in Table 2, Table 3 and Table 4, respectively. For the in vivo studies, a quantitative meta-analytical analysis was performed when possible (Table 5, Table 6, Table 7 and Table 8). By preference, the meta-analysis was conducted using studies where direct comparison was evaluated. However, when direct comparison was not available (or not enough), an indirect evaluation was performed. The two parameters quantitatively analysed in vivo were: (1) occurrence of severe inflammation (or necrosis/abscess formation) in the pulp tissue up to a period of 30 days (Table 5, Table 6 and Table 7), and (2) formation of a complete bridge between the material and the pulp tissue after 30 or more days (Table 8).

#### 3.3.1. Qualitative Analysis of In Vitro Studies

The qualitative analysis of the included in vitro studies showed that the materials that were studied the most in contact with human dental pulp cells of primary origin were the hydraulic calcium-silicate cements (hCSCs), Pro-Root MTA (Dentsply-Sirona; 9/26 studies) and Biodentine (Septodont; 7/26 articles) (Table 2). These were followed by MTA-Angelus (Angelus, Londrina, Brazil; 5/26 studies), the calcium-hydroxide (CH) cement Dycal (Dentsply-Sirona, Konstanz, Germany; 5/26 studies), the resin-free hCSC iRoot BP (Innovative Bioceramix; 4/26 studies), the resin-based calcium-silicate cement (Rb-CSC) Theracal LC (Bisco; 4/26 studies) and Portland cement (Ssangyong, Seoul, Korea; 4/26 studies). The other included materials were tested in three or less studies (Table 2). From the analysis, in general hCSCs (Pro-Root MTA, Dentsply-Sirona; Biodentine, Septodont; MTA-Angelus, Angelus; iRoot BP, Innovative Bioceramix; Portland cement, Ssangyong) were found to be non-cytotoxic when in (in)direct contact with human dental pulp cells, while they also exhibited bioactivity (migration, proliferation, mineralization capacities) towards human tooth pulp cells they were exposed to [28,59,61,62,63,65,66,67,68,70,74,77,80]. Moreover, the CH cement Dycal (Dentsply-Sirona) was in general deemed cytotoxic (in some studies cytotoxicity could not be tested because most of the cells died in contact with this calcium-hydroxide cement), and when directly compared, Dycal (Dentsply-Sirona) was more cytotoxic than the hCSCs tested [58,59,66,80]. The Rb-CSC Theracal LC (Bisco) seemed not to be as cytotoxic as Dycal (Dentsply-Sirona) [69], but when compared to hCSCs, contradictory results were reported as some authors showed similar results with both materials [70], while other authors showed more cytotoxicity and less bioactivity with Theracal LC (Bisco) [28,62].

#### 3.3.2. Meta-Analysis of the Effect of Pulp-Capping Agents on the Inflammatory Reaction Induced in Human Pulp Tissue

##### Inflammatory Reaction at Day 30

The inflammatory reaction induced by the different materials at day 30 is shown in Table 5. No significant difference was found in any of the direct or indirect comparisons/combinations tested. However, the quality/certainty of the evidence was very low or low following the GRADE recommendations. This was mostly due to the lack of studies directly comparing materials or the few studies available.

The materials that were tested the most were Pro-Root MTA (Dentsply-Sirona) and CH powder/saline—11 and eight studies, respectively. For these two materials, severe inflammation or necrosis/abscess formation were not reported in any of the included studies. Other studies investigated CH cements, such as Dycal (Dentsply-Sirona) or Life (Kerr, Orange, CA, USA); other MTA-like products, such as MTA-Angelus (Angelus); and resin-based adhesives.

##### Inflammatory Reaction at Day 15

The inflammation induced by the different materials at day 15 is shown in Table 6. No significant difference was found in any of the direct or indirect comparisons/combinations tested. However, not many studies were found in this time-period category and therefore not many comparisons could be included. In this case, the quality/certainty of the evidence was very low or low following the GRADE recommendations. This is mostly due to the lack of studies directly comparing materials or the few studies available.

##### Inflammatory Reaction up to Day 7

The inflammatory reaction induced by the different materials at day 7 is shown in Table 7. No significant difference was found in any of the direct or indirect comparisons/combinations tested. However, the quality/certainty of the evidence was very low or low following the GRADE recommendations. Likewise, as explained above for the 15 and 30-day periods, this must mostly be attributed to the lack of studies directly comparing materials or the few studies available.

##### Meta-Analysis of the Effect of Pulp-Capping Materials on Hard-Tissue Formation upon Capping Exposed Human Pulp Tissue

The ability of the each pulp-capping agent to induce dentin-bridge formation is shown in Table 8. Some comparisons between materials did reveal significant differences (Table 8). The most frequently tested materials were: (1) resin-free MTA-like cements within 14 studies [12 studies tested Pro-Root MTA (Dentsply-Sirona) and two studies Angelus-MTA (Angelus)], (2) calcium-hydroxide cements within 13 studies [nine studies tested Dycal (Dentsply-Sirona), three studies Life (Kerr), one study Calcimed (Cerkamed, Stalowa Wola, Poland) and one study Multi-Cal Liner (Pulpdent, Watertown, MA, USA)], (3) pure calcium-hydroxide powder/saline paste within 12 studies and (4) resin-based adhesives within 11 studies [three studies tested Scotchbond Multipurpose (3M, Seefeld, Germany), three studies Clearfil SE Bond (Kuraray Noritake, Tokyo, Japan), three studies Clearfil Liner Bond 2 (Kuraray Noritake, Tokyo, Japan), one study Single Bond (3M, Seefeld, Germany), one study Single Bond Universal (3M, Seefeld, Germany), one study All-Bond 2 (Bisco) and one study Prime&Bond NT 2.1 (Dentsply-Sirona, Konstanz, Germany)]. To have a better overview of how frequently the different capping agents were tested and their interactions, a graphic network analysis is shown in Figure 6.

The materials that more often induced dentin-bridge formation were calcium-hydroxide powder/saline and Pro-Root MTA (Dentsply-Sirona). Unfortunately, the two cements were not directly compared and the reported results originate from indirect comparisons (Table 8). However, when both materials were indirectly compared (Table 8), no significant difference in dentin-bridge formation was found (Relative risk (95% CI) = 1.64 (0.98, 2.77)), although the difference was statistically borderline non-significant in favour of calcium-hydroxide powder/saline (*p* = 0.06) (Figure 2). The quality/certainty of the evidence was ranked as moderate, since the data were gathered from many different studies conducted by many different authors (Table 8 and Figure 2). When MTA-Angelus (Angelus) was also taken into account, the difference between the materials was significant in favour of calcium-hydroxide powder/saline (relative risk (95% CI) = 2.24 (1.19, 4.20; *p* = 0.007); one study that directly compared materials was then included [52] (Table 8).

On the other hand, calcium-hydroxide cements, such as Dycal (Dentsply-Sirona) and Life (Kerr), performed significantly worse than calcium-hydroxide powder/saline (Relative risk (95% CI) = 3.23 (2.00, 5.20), *p* < 0.00001) and MTA-like cements (relative risk (95% CI) = 0.41 (0.23, 0.73), *p* < 0.0001) (Figure 3 and Figure 4, respectively).

When calcium-hydroxide cements were indirectly compared to hydraulic calcium-silicate cements (hCSCs), such as Biodentine (Septodont) and iRoot BP (Innovative Bioceramix, Vancouver, Canada), no significant difference was found in favour of hCSCs (relative risk (95% CI) = 0.47 (0.15, 1.44), *p* = 0.12) (Table 8). The difference was significant in favour of hCSC (*p* < 0.001) when the materials were directly compared. However, we found only one study where these two materials were directly compared, because of which this finding should be taken with caution [107]. The quality of evidence varied among the different comparisons from high (MTA-like materials vs. calcium-hydroxide (CH) cements), to moderate (CH cements vs. CH powder/saline) to low (hCSCs vs. CH cements) depending on the amount and quality of studies available.

Furthermore, the comparison between resin-based vs. resin-free materials was significantly different in favour of the resin-free materials (relative risk (95% CI) = 3.69 (2.23, 6.12), *p* < 0.0001) (Table 8 and Figure 5). The result was significantly different independently of which resin-free material was used (CH powder, CH cements, resin-free MTA-like cements or resin-free hCSCs).

### 3.4. Risk of Bias Across Studies

The risk of bias across studies (interpreted as the certainty of quality of evidence) was analysed using the GRADE recommendation. For most of the comparisons done between material categories for the parameter “Inflammatory reaction,” the certainty/quality in evidence was mostly evaluated as “very low” or “low.” This must mostly be ascribed to high risk of bias in individual studies but also to the few articles present in most categories increasing the imprecision and inaccuracy levels (Table 5, Table 6 and Table 7).

For the parameter “reparative bridge formation,” after pulp capping with different materials, most comparisons were categorized as of “moderate” or “high” quality. In this case, the number of articles that directly compared materials was higher than for the “inflammation” parameter. Moreover, the total number of cases available was also high, reducing the total imprecision and inaccuracy levels (Table 8).

## 4. Discussion

### 4.1. Summary of Results

The results of this systematic review showed that cell viability and bioactivity of the pulp-capping materials exposed to human dental pulp cells in vitro differ among the materials tested (Table 2). Therefore, the first null-hypothesis that there is no difference among the materials tested, was rejected.

Moreover, the outcomes of the meta-analysis regarding in vivo inflammatory effect of the different pulp-capping agents did not reveal differences among the materials tested (Table 5, Table 6 and Table 7). Therefore, we failed to reject the second null-hypothesis.

In vivo, the formation of a complete hard-tissue bridge in healthy human teeth after exposure and subsequent pulp capping varied as a function of the materials used (Table 8). In this way, the third null-hypothesis that there is no difference in hard-tissue formation among the materials tested, was also rejected.

The main conditions for a successful vital pulp therapy are: (1) a healthy patient with a good healing capacity, (2) a pulp-tissue environment free of bacteria and (3) a biocompatible material [45,114,115,116]. Of these three characteristics, we were interested in studying the effect of biocompatibility of the different pulp-capping materials on human pulp tissue. However, biocompatibility is a broad term requiring both in vitro and in vivo tests to be conducted. This is the reason why we included studies evaluating cytotoxicity and bioactivity of materials by exposing human pulp tissue in vitro and in vivo to these materials. There are also many studies evaluating biocompatibility towards pulp tissue in animals. However, it has been shown that there might be differences in reaction of pulp tissue from animals as compared to humans due to different metabolisms and immune system responses [117,118]. Therefore, because of the relatively high availability of human studies, we decided to include only studies performed on human dental pulp cells or human teeth.

The main reason for failure after vital pulp therapy is (re)infection of pulp tissue by bacteria [9,10,45,116,119]. Therefore, in order to selectively evaluate the effectiveness of different pulp-capping agents, we included only studies that performed vital pulp therapy on sound teeth from healthy volunteers using a strict aseptic environment (i.e., rubber dam isolation). By doing so, we intended to avoid cofounding factors like environmental contamination or lack of healing capacity of patients.

#### 4.1.1. Systematic Review of In Vitro Studies

The systematic review of the included in vitro studies showed that not much research has been done evaluating the direct effect of dental adhesives and resin composites on human dental pulp cells. Only two out of 26 studies (one for adhesives [28] and one for resin composites [78]) (Table 2) met the inclusion criteria (Table 1). The reason for this might be that the dental scientific community [based mostly on the results from in vitro studies and histology involving animal and human teeth (Table 3, Table 4 and Table 5)] has accepted that resins and monomers eluted from adhesives and composites are toxic for pulp cells [29,81,83,84,86,92,93,97,98,105,120,121,122,123,124,125]. However, this is in contradiction with the desire and increasing tendency from industry and researchers to develop resin-based pulp-capping materials, which are more user and patient friendly. Some examples of this kind of material are the resin-based calcium-hydroxide cements Prisma VLC (Dentsply-Sirona), Calcimol (Voco, Cuxhave, Germany) and Lime-Lite (Dentsply-Sirona), and the resin-modified calcium-silicate materials Theracal LC (Bisco) and Biocal Cap (Harvard).

It is difficult to make much out of the results of in vitro studies because most of them each use a different method to prepare the materials or to expose the cells. Moreover, not all of them refer to the ISO-standards (only 10/26 studies explicitly mentioned the use of ISO-standards) and the ISO-standards have been changing over time. This is important because if the data cannot be compared, the results are difficult to interpret and extrapolate. Furthermore, companies and researchers have to adhere to ISO-standards to develop/evaluate their materials. If the experiments are not done following these recommendations, the results might be less applicable.

Another particular issue with research involving hCSCs is that most researchers used “set” materials to conduct their experiments (only 2/26 included in vitro studies used “freshly-mixed” cements in their experimental protocol; Table 2) [66,76]. On the one hand, a protocol involving “set” materials is very useful, as it allows an easier set-up and more standardization of the experiments. However, care should be taken when designing the protocols, as hCSCs are very sensitive to drought (they need water to set and achieve their physio-mechanical properties), and, more importantly, in the clinic they are brought into contact with the pulp tissue when they are still setting (“freshly mixed”). Therefore, the use of “freshly-mixed” cements has a benefit in the case of these pulp-capping agents.

#### 4.1.2. Systematic Review and Meta-Analysis of In Vivo Studies

##### Inflammatory Reaction

From the analysis of the inflammatory reactions of human pulp tissue exposed to the biomaterials tested, we may conclude that there exists no difference in inflammatory reaction induced by the materials tested (Table 5, Table 6 and Table 7). However, we have to be cautious because evaluating inflammatory reaction can be very subjective. This is the reason why we decided to count as “*events*” only very strong inflammatory reactions, such as severe inflammation or necrosis/abscess formation. Even though in most of studies an independent examiner was selected to avoid risk of bias, the difference between slight and moderate inflammatory reaction is very subjective. Severe inflammation or necrosis are in principle more objective outcomes. We should also bear in mind that hard-tissue histology, particularly tooth histology, is a difficult technique, even for pathologists (as they are not often confronted with hard tissues such as teeth). Much methodological experience is needed before quality sections can be prepared and trustable information gathered. It may happen that artefacts caused by incorrect fixation or induced during demineralization and cutting procedures are confused with pulp-tissue damage. Moreover, in most studies a limited number of sections are obtained. This is not ideal, as pulp tissue may appear normal in one part but inflamed in another part. Therefore, it is recommended that the whole pulp tissue is cut and sections from different areas are stained and analysed to guarantee representativeness [126].

##### Complete Reparative Bridge Formation

For the in vivo quantitative analysis, different comparisons were conducted. Comparison between resin-based and resin-free materials clearly elicited that the latter produce more frequently complete dentin bridges than the former (*p* < 0.001; Table 8 and Figure 5). Comparison of Pro-Root MTA (Dentsply-Sirona) with calcium hydroxide (CH) yielded contradictory results. When we analysed studies comparing pure CH powder with Pro-Root MTA (Dentsply-Sirona), pure CH powder produced more frequently complete bridges than Pro-Root MTA (Dentsply-Sirona), although no statistically significant difference was recorded (*p* = 0.06; odds ratio (95% CI) = 1.85 (0.98, 3.49)) (Figure 2). However, when we evaluated studies comparing CH cements, such as Dycal (Dentsply-Sirona) vs. Pro-Root MTA (Dentsply-Sirona), a significant difference in the formation of complete bridges was found in favour of MTA (*p* < 0.01; Table 8 and Figure 3). This is relevant, especially in places were the high cost and/or lack of a national insurance system make it difficult for clinicians and patients to afford materials like Biodentine (Septodont) or Pro-Root MTA (Dentsply-Sirona). Moreover, many researchers and clinicians consider CH cements, such as Dycal (Dentsply-Sirona) and Life (Kerr), to be the same as or similar to as pure CH powder, and therefore, as *gold-standard* materials for vital pulp therapy [69]. This has led to many clinical trials evaluating pulp-capping treatment with CH cements serving as control (“gold-standard”) materials, with disastrous results [7,16,17,127,128]. Therefore, as previous research had already highlighted, and considering the results of this meta-analysis, pure CH powder and CH cements should not be considered to be the same [52,129,130]. Even though the quantity/quality of mineral bridge formation seems higher when using resin-free hCSCs [131], pure CH powder is a very cheap alternative with excellent efficiency for vital pulp therapy. Its efficacy has been backed up with almost 100 years of scientific evidence behind it, including in vitro, ex vivo and in vivo studies on animal and human teeth [126,132,133,134].

### 4.2. Comparison with Previous Studies and Limitations

Recently, there have been many review articles about this topic [48,49,50,51,135]. However, some limitations of these reviews were that, (1) they focused on the comparison between CH compounds and MTA [50]; (2) they only included in their analyses studies with a direct comparison between the materials tested [48,49] or (3) they only compared in vitro articles [51,135]. In our systematic review, we wanted to go further, and we included all types of pulp-capping materials and, both in vitro and in vivo studies. Moreover, we also did an indirect comparison between the different materials tested, similarly to a network meta-analysis [136]. By doing so, we were able to include a comparison between pure CH powder and Pro-root MTA (Dentsply-Sirona), as there has been, to our knowledge, no research performed in human teeth comparing both materials directly [112,137].

Some of the limitations of this study are the narrow scope of the review by including only in vitro studies performed on human dental pulp cells from primary sources and in vivo studies performed on healthy permanent human teeth using strict aseptic criteria. However, we think that there have been other recent reviews studying the effect of these type of materials on other cell lines (i.e., human dental (stem) cells from the apical papilla (SCAPs) or (stem) cells from human exfoliated deciduous teeth (SHEDs)) and also using animal cell lines/tissue or teeth [48,135]. Other limitations may be the limited timespan of the review, as we took into consideration only articles from 1993, and the short-term follow-up of the in vivo studies. Regarding the timespan of the review, we did so because the first hydraulic calcium-silicate cement (Pro-Root MTA, Dentsply-Sirona) was developed in 1993 and the first article was published in 1995. Moreover, the use of human dental pulp cells of primary origin became popular from the year 2000 and on, after publication that stem cells could be harvested from human teeth [138,139,140]. In relation to the short-term follow-up of the in vivo studies, we wanted to evaluate the histological characteristics of the inflammatory reaction and the reparative bridge formation after exposure with different pulp-capping agents. In the long-term, other factors like the health status of the patient or the quality of the coronal restoration play important roles. Therefore, in order to exclude or minimize these co-founding factors, we wanted to focus solely in the (short-term) histological features of the healing after pulp-capping. However, care should be taken when interpreting the results of this systematic review and meta-analysis, as some articles studying the effects of dental materials on human cells or teeth might have been left aside for those reasons.

## 5. Conclusions and Recommendations for Future Research

In conclusion, we want to stress that materials such as pure calcium-hydroxide (CH) powder and Pro-Root MTA (Dentsply-Sirona) have shown excellent biocompatibility in vitro and in vivo when tested on human cells and teeth. Their use after many years of research and clinical experience seems safe and proven for vital pulp therapy in healthy individuals given that an aseptic environment (rubber dam isolation) is maintained. More recently introduced hCSCs like Biodentine (Septodont) and iRoot BP (Innovative Bioceramix) showed promising results, but more studies are necessary to confirm their suitability. However, in general, all these hCSCs have many disadvantages, such as long setting times, lack of bonding capacity to tooth structures, high solubility and high risk of discoloration, which make them difficult to handle and less user and patient friendly.

Newer, setting-on-command materials, such as resin-based calcium-silicate cements, are therefore highly desirable. Nevertheless, before these innovative materials can be used in patients, improved bioactivity and biocompatibility are needed and mandatory. Moreover, for these materials to be successful in the clinical practice, better bonding capacity to hard dental tissues and enhanced physio-mechanical properties are also needed. On the one hand, by having a strong bonding ability to dentin, better sealing ability and reduced long-term bacterial leakage is expected. On the other hand, increased physio-mechanical properties may ensure that these materials can be used also as definitive restorations. We highly encourage clinicians not to use new materials in patients until sufficient scientific evidence has been provided. Even minor changes in composition may have drastic consequences in the clinical outcomes of our treatments.

## Figures and Tables

**Figure 1 materials-13-02670-f001:**
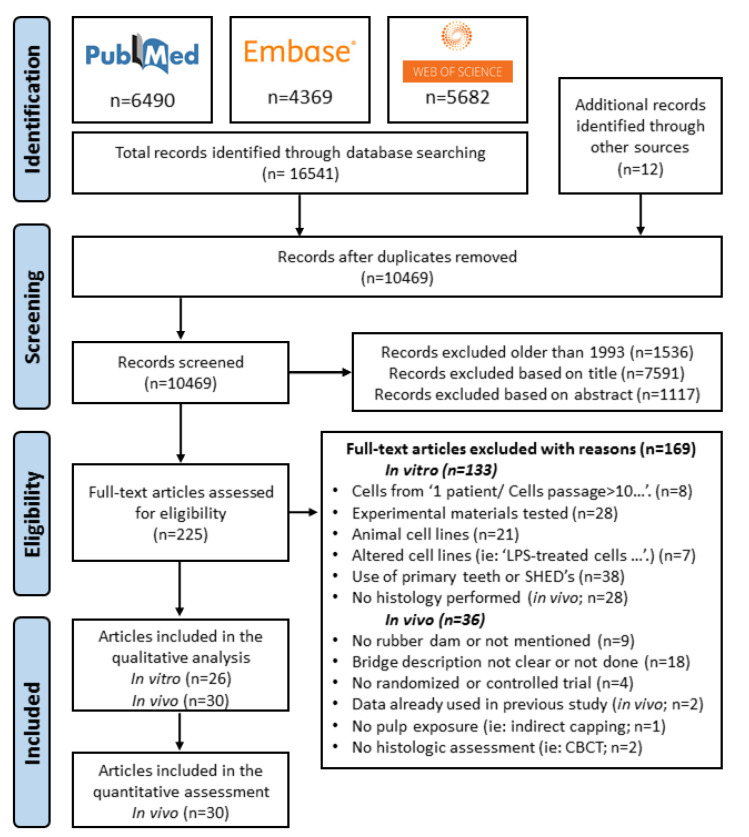
PRISMA flowchart describing the article screening procedure.

**Figure 2 materials-13-02670-f002:**
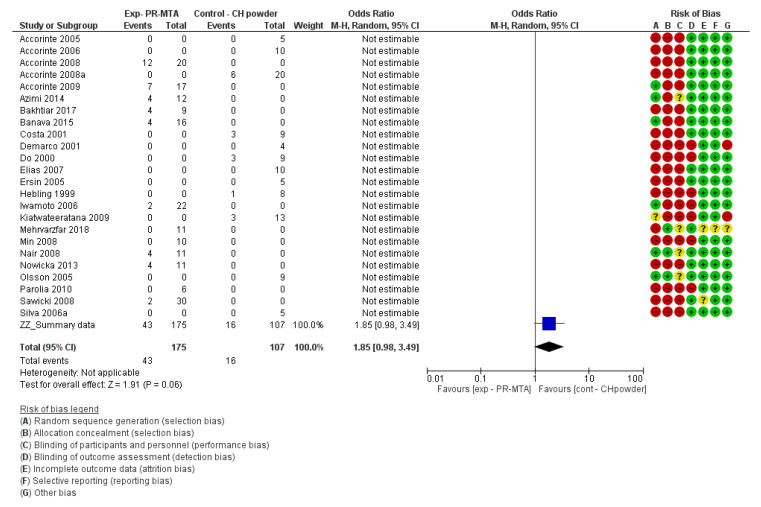
Forest plot and risk of bias of the studies evaluating the formation of a complete reparative bridge comparing calcium-hydroxide powder (CH powder) with Pro-Root MTA (Dentsply-Sirona) (odds ratio (95% CI), random effects). An “*event*” was considered the lack of a complete hard bridge after 30 days of pulp-capping.

**Figure 3 materials-13-02670-f003:**
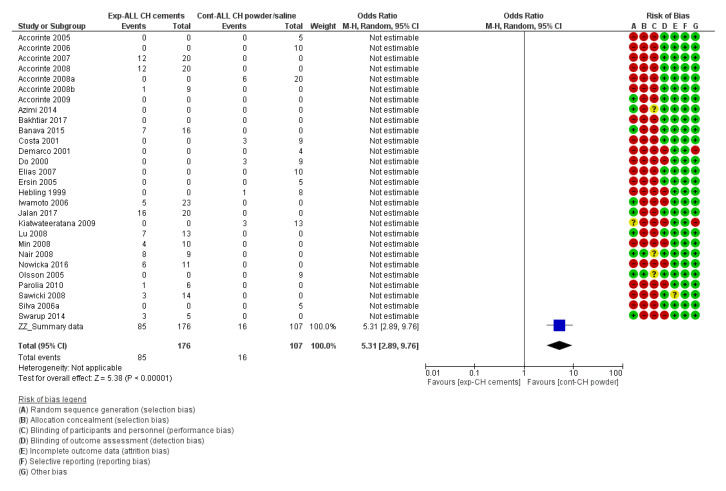
Forest plot and risk of bias of the studies directly comparing the formation of a complete reparative bridge when calcium-hydroxide (CH) cements or CH powder/saline paste were used (odds ratio (95% CI), random effects). An “*event*” was considered the lack of a complete hard bridge after 30 days of pulp-capping.

**Figure 4 materials-13-02670-f004:**
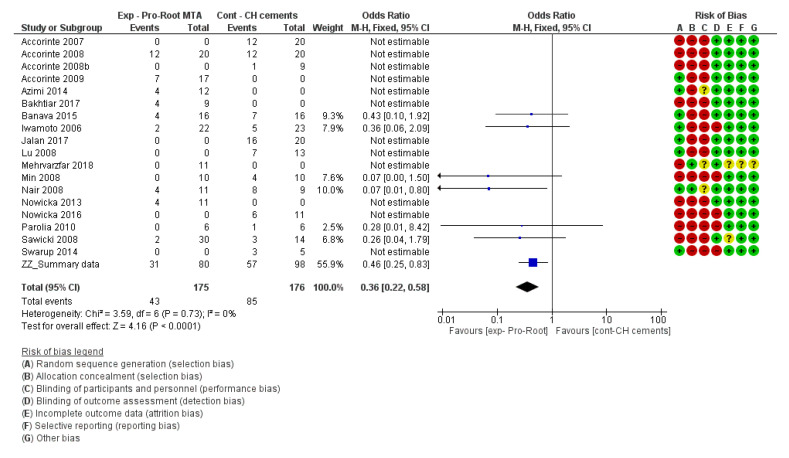
Forest plot and risk of bias of the studies directly comparing Pro-Root MTA (Dentsply-Sirona) to calcium-hydroxide cements (CH cements) for the formation of a complete reparative (odds ratio (95% CI), fixed effects). An “*event*” was considered the lack of a complete hard bridge after 30 days of pulp-capping.

**Figure 5 materials-13-02670-f005:**
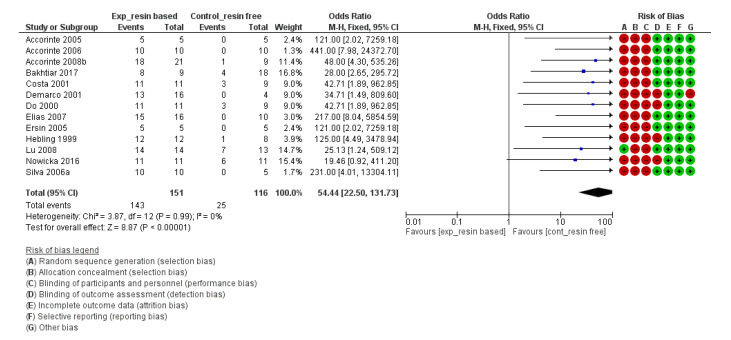
Forest plot and risk of bias of the studies directly comparing the formation of a complete reparative bridge with resin-based and resin-free materials (including pure calcium hydroxide powder, calcium hydroxide cements and hydraulic Calcium-Silicate cements) (odds ratio (95% CI), random effects). An “*event*” was considered the lack of a complete hard bridge after 30 days of pulp-capping.

**Figure 6 materials-13-02670-f006:**
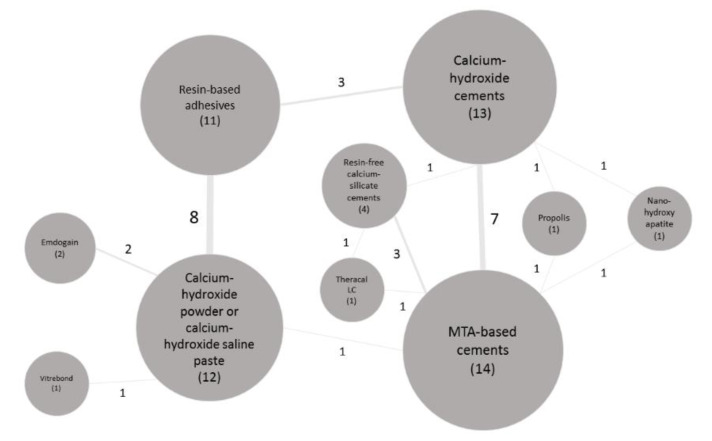
Schematic representation (network) of the interaction among the in vivo studies (n = 30). The balls (nodes) represent the materials and the times each material was studied. The line thickness and the number connected to the lines (edges), linking two materials, represent the frequency of interactions between them.

**Table 1 materials-13-02670-t001:** Eligibility criteria for in vitro and in vivo studies.

**Eligibility criteria for in vitro studies**		
**Characteristics**	**Inclusion Criteria**	**Exclusion Criteria**
Publication year	Studies published from 1993	Studies published before 1993
Language	English	Other languages different than English
Population	Human dental pulp cells from a primary source Cells from more than 1 patient	Other type of oral/dental cells or cells not from primary source (i.e.,: immortalized cells, commercial cell lines, …) Cells obtained from one single patient
Tests performed	At least 2 different tests were performed	Only 1 test was performed (i.e.,: cytotoxicity, mineralization ability, ...)
Materials tested	Commercially available materials Resin-based or resin-free pulp-capping materials	Experimental materials or modification of an existing material Materials for other purposes (sealers, fixation cements, monomers, …)
Analysis	At least 2 different materials compared between each other	Materials compared only to the control
**Eligibility criteria for in vivo studies**		
Publication year	Studies published from 1993	Studies published before 1993
Language	English	Other languages different than English
Ethical Committee	Mentioned	No ethical committee mentioned in the text
Population	Permanent teeth Healthy teeth (free from caries or infection) Patients’ age and amount of teeth should be provided	Primary teeth Teeth exhibiting caries or periodontal disease Sample size and age of the participants not mentioned
Materials tested	Commercially available materials Resin-based or resin-free pulp-capping materials Clear description and brand of the materials used	Experimental materials or modification of an existing material Materials for other purposes (sealers, fixation cements, monomers, …) No clear description and brand of the materials used
Analysis	Histology Inflammation and hard-tissue formation	Studies where “only” clinical parameters were evaluated No bridge formation evaluated
Type of study	Randomized or controlled clinical trials (RCT or CCT)	Other type of studies (case reports, case series, …)

**Table 2 materials-13-02670-t002:** Included in vitro studies. Symbols of greater than (>), less than (<) or equal to (=) are used to compare the results of the tested groups.

**Studies**	**Materials**	**Type of Exposure**	**Parameters**	**Methods**	**Results**
Alliot-Licht et al. (1994) [56]	Calcium hydroxide (CH) Hydroxyapatite (HAp)	CH particles sterilized by heating (180 °C-1 h); direct contact (materials powder in culture medium)	Cell morphology	Light microscopy (at 3 & 5 days)	CH inhibited pulp fibroblasts growth (<cell density than control; subjective observation) HAp did not affect the cell density (≈cell density as the control; subjective observation)
			Phagocytotic activity	SEM (at day 5)	Close contact of CH particles with fibroblasts’ membrane. HAp particles were closely bound to cell membrane or internalized by the cells.
				TEM (at day 5)	Cells cultured in the presence of CH exhibited ghost cells and electron-dense spherical vesicles in the cytoplasm of living cells. TEM revealed HAp particles within the cells.
			Cell proliferation	DNA synthesis (at 1, 2, 3 & 4 days)	CH and HAp delayed the proliferation of cells at all time points.
				Protein synthetic activity (at 6 days)	CH < incorporation of [^3^H]-leucine and [^3^H]-proline by pulp fibroblasts at day 6. HAp > incorporation of [^3^H]-leucine and [^3^H]-proline by the pulp fibroblasts at day 6.
			Cell differentiation	ALP activity (at 8 days for CH; at 5 & 8 days for HAp)	CH inhibited ALP activity of pulp fibroblasts at day 8. HAp inhibited ALP activity of pulp fibroblasts at 5 and 8 days.
Min et al. (2007) [57]	Portland cement (PC) Portland cement with bismuth oxide (BPC)	Indirect contact (SET materials)	Cell viability	MTT assay (at 12, 24, 48 & 72 h)	PC > BPC at 12 and 24 h PC ≈ BPC at 48 and 72 h
			Nitric oxide production	Griess reaction(at 12, 24, 48 & 72 h)	BPC > nitrite production than PC at 12 and 24 h. PC ≈ BPC nitrite production at 48 and 72 h.
			Ho-1 and iNOS	RT-PCR (at 12, 24, 48 & 72 h)	Ho-1: PC < BPC at all study periods iNOS: PC < BPC at all study points
Min et al. (2007) [58]	Portland cement (PC) Fuji-II LC (Fuji-II, GC) Zinc-oxide Eugenol (IRM; Dentsply-Sirona) CH cement Dycal (Dentsply-Sirona)	Direct and indirect contact (SET materials)	Cell morphology	SEM (at 24 h)	PC: showed flattened cells close to one another and spreading across the substrate. Fuji-II, IRM, and Dycal: no living cells were seen.
			Cell viability	MTT assay (at 12, 24, 48 & 72 h)	PC ≈ control at all study periods. PC > Fuji-II, Dycal and IRM at all study periods. Control > Fuji II LC, IRM, and Dycal at all study points.
			Cell differentiation	RT-PCR (ON, DSPP) (at 7 days)	ON: PC ≈ positive control group. DSPP: PC stimulated mineralization but less than the positive control.
Laurent et al. (2008) [59]	Ca_3_SiO_5_ cement (CS) Dycal (Dentsply-Sirona) Pro-Root MTA (MTA; Dentsply-Sirona)	Indirect contact * (SET materials) [ISO-Standard–(Nr. Not mentioned)]	Cell viability	MTT assay (at 24 h)	No contact (disk diffusion): CS ≈ MTA ≈ Dyc Indirect contact (eluates from materials): Cs ≈ MTA > Dycal
			Cell differentiation	Immunohistochemistry (at 4 weeks)	MTA and CS expressed Nestin and Collagen I at a similar level as the control group. Both materials generated mineral deposits at a similar level as the control group.
			Genotoxicity	Ames test	CS does not induce reverse mutations with/without the S9 metabolic activation system.
				Micronuclei test Comet assay	CS generated lymphocytes with micronuclei ≈ as the negative control.
					CS generated DNA in the tail ≈ as the negative control and < than the cytotoxic control.
Min et al. (2009) [60]	Radiopaque Portland cement (RPC) Portland cement (PC) IRM (Dentsply-Sirona)	Direct and indirect contact (SET materials)	Cell morphology	SEM (at 48 h)	PC and RPC: Spread and flattened HDPCs. The density and characteristics of the HDPCs in both groups were similar to that on control samples. IRM: no living cells were seen in contact with the
			Cell differentiation	ALP activity (at 1, 3, 7 & 14 days)	1d: PC and RPC > control; 3d, 7d, 14d: control > PC and RPC 2wk and 3wk: PC and PCR > control DSPP: PC and RPC > control at day 14; OCN: control ≈ Pc and RPC at all study periods.
				ARS staining (1, 2 & 3 wk)	
				RT-PCR (DSPP, ON) (at 1, 3, 7 & 14 days)	
Lee et al. (2014) [61]	ProRoot MTA (MTA; Dentsply-Sirona) α-tricalcium phosphate-based cement (α-TCP)	Direct and indirect contact (SET materials)	Cell morphology	SEM (at 72 h)	hDPCs in contact with MTA and α-TCP were well-spread and flattened.
			Cell viability	MTT assay (at 1, 2, 3, 7 & 14 days)	MTA and α-TCP ≈ control until day 7 α-TCP > MTA at 14d; α-TCP ≈ control
			Cell differentiation	Western blot (DSPP, DMP-1 and ON) (at 3 days)	α-TCP ≈ MTA for DSPP, DMP-1 and ON.
				ARS staining (at 14 days)	α-TCP ≈ MTA for DSPP, DMP-1 and ON.
				Immunofluorescence (DSPP, DMP-1 and ON) (at 7 days)	α-TCP and MTA induced higher protein signals than the control group.
Bortoluzzi et al. (2015) [62]	Biodentine (Bd; Septodont) Theracal LC (Th; Bisco) MTA Angelus (MTA-A; Angelus)	Indirect contact * (SET materials)	Cell viability	XTT assay (direct and indirect eluate evaluations) Flow cytometry– Annexin V-PI (4 weekly cycles)	Direct evaluation: 1st cycle: control > Bd > MTA-A and Th 2nd cycle: control > Bd ≈ MTA-A > Th 3rd cycle: control ≥ MTA-A ≥ Bd > Th 4th cycle: control ≈ Bd ≈ MTA-A > Th Indirect eluate evaluation: 1:1&1:10 dilutions: control > MTA-A ≈ Bd > Th; 1:100 dilution: control ≈ MTA-A ≈ Bd > Th Percentage of healthy, non-apoptotic and non-necrotic cells: control > MTA-A ≈ Bd > Th Th was the most cytotoxic material causing apoptosis and necrosis.
			Cell differentiation	qRT-PCR (DSPP, OCN, BSP, RUNX 2, DMP-1 and ALP) (at 7 days)	ALP; OCN; BSP; DSPP; DMP-1: Bd and MTA-A > control ≈ Th RUNX 2: Bd ≈ MTA-A ≈ control ≈ Th
				ALP activity (at 14 days)	Bd ≈ control > MTA-A > Th
				ARS and TEM (at 21 days)	Bd > control > MTA-A > Th
Niu et al. (2015) [63]	ProRoot MTA (MTA; Dentsply-Sirona) Quick-Set2 (Qs; Avalon Biomed Inc)	Direct and indirect contact (SET materials)	Cell viability	Flow cytometry– Annexin V-PI (3 weekly cycles) Leakage of cytosolic enzyme (3 weekly cycles) Caspase-3 acitivity (3 weekly cycles) Oxidative stress (3 weekly cycles)	Number of healthy cells: 1st cycle: control > Qs > MTA (*p* < 0.001) > IRM 2nd cycle: control > Qs ≈ MTA > IRM; 3rd cycle: control ⩾ MTA ⩾ Qs > IRM Percentage of cytotoxicity: 1st cycle: IRM > MTA > Qs > control; 2nd and 3rd cycles: IRM > MTA ≈ Qs > control Relative caspase-3 activity: 1st cycle: IRM > MTA > Qs > control; 2nd cycle: IRM > MTA > Qs > control 3rd cycle: IRM > MTA ≈ Qs > control Oxidative stress: 1st cycle: IRM > MTA > Qs > control; 2nd cycle: IRM > MTA ≈ Qs > control 3rd cycle: IRM > MTA ≈ Qs ≈ control
			Cell proliferation	MTT assay (3 weekly cycles)Cellular DNA content (3 weekly cycles)	1st cycle: control > Qs > MTA > IRM 2nd cycle: control > Qs ≈ MTA > IRM 3rd cycle: control > Qs > MTA > IRM DNA content: 1st cycle: control > Qs > MTA > IRM 2nd cycle: control > Qs ≈ MTA > IRM 3rd cycle: control > Qs ≈ MTA > IRM
Öncel Torun et al. (2015) [64]	iRoot BP Plus (iBP; Innovative Bioceramix) White MTA Angelus (MTA-A; Angelus)	Indirect contact (SET materials)	Cell viability	XTT assay (24, 48 & 72 h)	24 h; 1:1 and 1:2 dilutions: iBP > W-MTA-A; 1:5 and 1:10 dilutions: iBP ≈ MTA-A 48 h; 1:1 dilution: iBP > W-MTA-A; 1:2, 1:5 and 1:10 dilutions: iBP ≈ MTA-A 72 h; all concentrations: iBP ≈ MTA-A
			Cell differentiation	qRT-PCR (BMP-2, ON, BSP, OPN, DSPP, Col I A1, HO-1 at 24 & 72 h)	BMP-2: 24 h 1:1 and 1:5 dilutions MTA-A > iBP; 1:2 dilution: iBP ≈ MTA-A 72 h 1:1, 1:2 and 1:5 dilutions MTA-A > iBPON: 24 h 1:1 and 1:5 dilutions iBP > MTA-A; 1:2 diution: iBP ≈ MTA-A 72 h: 1:1 dilution iBP > MTA-A; 1:2 and 1:5 dilutions MTA-A > iBP BSP: 24 h: 1:1 dilution MTA-A > iBP; 1:2 and 1:5 diutions: iBP ≈ MTA-A 72 h: 1:1 and 1:2 dilutions MTA-A > iBP; 1:5 diution: iBP ≈ MTA-AOPN: 24 h: 1:2 dilution iBP > MTA-A; 1:1 and 1:5 dilutions: iBP ≈ MTA-A 72 h: 1:1 and 1:5 dilutions MTA-A > iBP; 1:2 dilution: iBP ≈ MTA-A DSPP: 24 h: 1:1 dilution iBP > MTA-A; 1:2 and 1:5 dilutions: iBP ≈ MTA-A 72 h: 1:2 dilution iBP> MTA-A; 1:1 dilution MTA-A > iBP; 1:5 dilution: iBP ≈ MTA-ACol I A1: 24 h: 1:1 dilution iBP > MTA-A; 1:2 and 1:5 dilutions: iBP ≈ MTA-A 72 h: 1:1 and 1:2 dilutions iBP > MTA-A; 1:5 dilution: iBP ≈ MTA-AHO-1: 24 h 1:1 and 1:2 dilutions MTA-A > iBP; 1:5 dilution: iBP ≈ MTA-A 72 h: 1:1, 1:2 and 1:5 dilutions MTA-A > iBP
Zhang et al.(2015) [65]	iRoot BP Plus (iBP; Innovative Bioceramix) ProRoot MTA (MTA; Dentsply-Sirona)	Indirect contact (SET materials) (ISO 10993-5)	Cell Viability	Flow cytometry– Annexin V-PI	iBP ≈ MTA ≈control
			Cell Migration	Wound-healing at 24 h	iBP ≈ MTA > control
				Transwell assay at 24 h	iBP = MTA > control
			Cellular adhesion and motility	Western-Blot (at 5, 10, 30 & 60 min) Cell Immunofluorescence assay (at 1 h)	iBP led to phosphorylation of p38 MAPK, ERK 1/2, JNK, Akt, and FGFR
					iBP significantly increased p–focal adhesion kinase (p-FAK), p-paxillin, and vinculin Cells treated with iBP showed highly organized and stretched stress fiber assembly
Chung CJ et al. (2016) [66]	Dycal (Dy; Dentsply-Sirona) Endocem Zr (E-Zr, Maruchi) White ProRoot MTA (MTA; Dentsply-Sirona) Retro-MTA (R-MTA; Bio MTA)	Indirect and direct contact; SET (s) and FRESH (f) materials	Cell morphology/attachment	Phase microscopy (at 3 & 7 days) SEM (at 3 & 7 days)	3d: MTA > cell morphology and attachement than R-MTA and E-Zr 7d: MTA, R-MTA and E-Zr sowed good cell morphology and attachement Dycal treated cells were dead after 3 and 7 days. Dycal was not further used
			Cell viability	XTT assay (at 3 & 7 days)	3 d: control ≈ MTA (s) ≈ MTA (f) > R-MTA (s) ≈ R-MTA (f) > E-Zr (s) ≈ E-Zr (f) 7 d: MTA(f) > control ≈ MTA (s) ≈ R-MTA (s) ≈ R-MTA (f) ≈ E-Zr (f) > E-Zr (s)
			Angiogenic properties	ELISA (VEGF, angiogenin, FGF-2) (at 3 & 7 days)	VEGF24 h: control ≈ MTA (s) ≈ R-MTA (s) ≈ R-MTA (f) ≥ MTA (f) ≈ E-Zr (s) > E-Zr (f) VEGF72 h: MTA (s) ≈ MTA (f) ≈ R-MTA (s) ≈ E-Zr (s) ≥ control ≥ R-MTA (f) ≈ E-Zr (f) Angiogenin 24 h: control≈ MTA(s) ≥ R-MTA (s) > MTA(f) > R-MTA (f) ≈ E-Zr(s)>E-Zr (f) Angiogenin 72 h: R-MTA (s) ≈ R-MTA (f) ≈ control > MTA(s) ≈ MTA (f) > E-Zr (s) ≈ E-Zr (f) FGF-2 24 h and 72 h: no difference among materials and control
Daltoé M et al. (2016) [67]	Biodentine (Bd; Septodont) White ProRoot MTA (MTA; Dentsply-Sirona)	Indirect contact (SET materials) (ISO 10993-5)	Cell Viability	MTT assay (at 24 & 48 h)	24 h: control ≈ MTA_1:100 ≈ Bd_1:100 > MTA_1:10, Bd_1:10, Bd_1:1 and MTA_1:1 48 h: control ≈ MTA_1:100 ≈ Bd_1:100 > MTA_1:10, Bd_1:10, MTA_1:1 and Bd_1:1
			Cell differentiation	qRT-PCR (SPP1, IBSP, DSPP, ALP 1, DMP-1 and RUNX 2 (at 24 & 48 h)	SPP1 & ALP1 & RUNX2 at 24 h: Bd and MTA ≈ control SPP1 & ALP1 & RUNX2 48 h: Bd and MTA > control IBSP & DSPP & DMP1: 24 h and 48 h: no expression
Widbiller M et al. (2016) [68]	Biodentine (Bd; Septodont) GI Ketac-Molar (KM; 3M) ProRoot MTA (MTA; Dentsply-Sirona)	Indirect and direct contact (SET materials)	Cell morphology/attachement (only Bd)	SEM (at 24 h)	Biodentine: cells showed adhesion to and spreading onto the cement surface * Not done for the other materials.
			Cell viability	MTT assay (at 1, 3, 5, 7, 10 & 14 days)	Bd > other materials and control at 1, 3, 5 and 7 d; Bd ≈ MTA > control > KM at 10 and 14 d MTA < viability than the control at 1d; MTA ≈ cell viability as the control at 3-5-7d; KM < cell viability than all the materials tested and the control at all time points
			Cell differentiation Not performed on KM	RT-qPCR (ALP, Col-I A1, DSPP, RUNX 2) (at 7, 14 & 21 days)	Col-I A1 & ALP: upregulated at 7d, especially for MTA, and decreased steadily until 21d DSPP: upregulated for MTA and BD at 14 and 21d RUNX2: downregulated for MTA and BF throughout the whole study period
				ALP activity (at 3, 7 & 14 days)	ALP activity was downregulated for Bd at all times: MTA ≈ control > Bd
Jeanneau C et al. (2017) [28]	Biodentine (Bd; Septodont) Theracal LC (Th; Bisco) Xeno III (Dentsply-Sirona)	Indirect contact (SET materials)	Cell proliferation	MTT assay (3, 5, & 7 days)	Bd_0.05 cm^2^/mL > Th_ 0.05 cm^2^/mL (*p* < 0.05) ≈ control at 3, 5 and 7 days Bd_0.5 cm^2^/mL > Th_ 0.5 cm^2^/mL (*p* < 0.05) ≈ control at 3, 5 and 7 days
			Cell differentiation	Immunofluorescence (DSP and Nestin at day 7)	Bd increased the expression of both markers, while Th had no effect
			Inflammatory effect	ELISA (IL-8; 24 and 48 h)	IL-8 expression at 24 h: Th_0.05 cm^2^/^mL^ > Bd_0.05 cm^2^/^mL^ ≈ control IL-8 expression at 48 h: Th_0.05 cm^2^/^mL^ > Bd_0.05 cm^2^/^mL^ > control
Jun S-K et al. (2017) [69]	Activa Bioactive (Activa; Pulpdent) Dycal (Dy; Dentsply-Sirona) Theracal LC (Th; Bisco)	Indirect contact (SET materials) (ISO 10993-12)	Cell viability	MTS assay (at 24 h) Live/dead assay (at 24 h)	3.125% eluates: Dy > Th > Activa ≈ control; 6.25% eluates: Dy > Th ≈ Activa ≈ control 12.5% eluates: Dy ≈ Th ≈ Activa ≈ control; 25% eluates: Dy < Activa < Th < control 50% eluates: Dy ≈ Activa < Th < control 50% eluates: Dy < Activa < Th < control
			Cell differentiation	ALP (at days 14 and 21)	14 d: Th > Dy > Activa ≈ Osteogenic medium 21 d: Th ≈ Dy > Activa > Osteogenic medium
				ARS (at 21 days)	Th ≈ Dy > Activa ≈ Osteogenic medium
Lee B-N et al. (2017) [70]	ProRoot MTA (MTA; Dentsply-Sirona) Theracal LC (Th; Bisco)	Indirect contact (SET materials)	Cell viability	WST-1 assay (at 24 h)	100% concentration: Th > MTA; At 50%, 25% and 10% dilutions: Th ≈ MTA At 100% MTA: cell viability < 70% and significantly lower than Th.
			Cell differentiation	RT-PCR (DSPP, DMP-1 at 1 & 3d) Q-PCR(DSPP, DMP-1 at 2, 5 & 7d) ALP staining (at day 7) ARS (at day 14)	DSPP 1 d: MTA > Th ≈ control; 3 d: MTA ≈ Th > control DMP-1 at 1 and 3d: MTA ≈ Th ≈ control DSPP & DMP-1: upregulated for both materials, especially at day 7. MTA > Th ≈ control MTA > Th > control
Mestieri LB et al. (2017) [71]	White MTA Angelus (MTA-A; Angelus) White Portland Cement (PC; Votoran)	Indirect contact (SET materials)	Cell viability	MTT assay	1:2, 1:3, 1:4 and 1:8 dilutions: control > W-MTA-A >W-PC 1:6 dilution: MTA-A ≈ control > PC
				Trypan blue assay	1:2 dilution: control > MTA-A > PC 1:3 dilution: control > PC > MTA-A 1:4 and 1:6 dilutions: control > PC > MTA-A 1:8 dilution: control > PC ≈ MTA-A
			Cell Differentiation	ALP activity (at 1, 3 & 7d)	1, 3 and 7 d: MTA-A ≈ PC ≈ control
Rodrigues EM et al. (2017) [72]	MTA-Plus (MTA-P; Prevest Denpro) White MTA Angelus (MTA-A; Angelus)	Indirect contact (SET materials)(ISO-10993)	Cell viability	MTT Assay Flow cytometry– Annexin V-PI	1:2 concentration: MTA-P > MTA-A ≈ control 1:4 and 1:8 concentrations: MTA-P ≈ MTA-A > control MTA-A > live cells than MTA-P ≈ control MTA-A > necrotic cells than MTA-P > control
			Cell differentiation	ALP activity (at 1, 3 & 7 days)	MTA-A < control < MTA-P after 7 days.
				ARS (14d)	MTA-A > MTA-P > control
				qRT-PCR (BMP2, OC, ALP)	Day 1_BMP2 & OC: MTA-A > MTA-P > control; ALP: MTA-A ≈ MTA-P < control Day 3_ BMP2: MTA-A > MTA-P > control; OC & ALP: MTA-A ≈ MTA-P < control
Sun Y et al. (2017) [73]	Biodentine (Bd; Septodont) iRoot FS (iFS; Innovative Bioceramix)	Indirect contact (SET materials)	Cell proliferation	CCK-8 assay (1, 3 & 7 days)	1 d: Bd_0.2 mg/mL ≈ Bd_2 mg/mL ≈ iFS_0.2 mg/mL ≈ iFS_2 mg/mL ≈ control (*p* ≥ 0.05) 3 d: Bd_0.2 mg/mL ≈ Bd_2 mg/mL ≈ iFS_0.2 mg/mL ≈ iFS_2 mg/mL > control 7 d: Bd_0.2 mg/mL ≈ iFS_0.2 mg/mL > Bd_2 mg/mL ≈ iFS_2 mg/mL > control
			Cell migration (24 h)	Wound healing assay Transwell migration assay	iFS_0.2 mg/mL > iFS_2 mg/mL > control > Bd_0.2 mg/mL > Bd_2 mg/mL
					iFS_0.2 mg/mL > iFS_2 mg/mL > control > Bd_0.2 mg/mL > Bd_2 mg/mL
			Cell differentiation	ALP activity (at 7, 14 d)	7 d: iFS_0.2 mg/mL ≈ iFS_2 mg/mL ≈ Bd_0.2 mg/mL > Bd_2 mg/mL > control
					14 d: iFS_0.2 mg/mL > Bd_0.2 mg/mL > Bd_2 mg/mL ≈ iFS_2 mg/mL > control
				ARS (at 21 d)	21 d: iFS_0.2 mg/mL > Bd_0.2 mg/mL ≈ Bd_2 mg/mL ≈ iFS_2 mg/mL ≈ control
				qRT-PCR (Col I and OCN) (at 1, 7 & 14 d)	1 d: Col I control ≥ all materials OCN iFS_2 mg/mL ≥ iFS_0.2 mg/mL ≈ Bd_0.2 mg/mL ≈ Bd_2 mg/mL ≈ control 7 d: Col I control > iFS_0.2 mg/mL > Bd_0.2 mg/mL > Bd_2 mg/mL > iFS_2 mg/mL OCN iFS_0.2 mg/mL > control ≈ iFS_2 mg/mL > Bd_2 mg/mL ≥ Bd_0.2 mg/mL
					14 d: Col I iFS_0.2 mg/mL > Bd_0.2 mg/mL ≈ Bd_2 mg/mL ≥ control ≥ iFS_2 mg/mL OCN iFS_0.2 mg/mL ≈ iFS_2 mg/m ≥ control ≥ Bd_0.2 mg/mL ≥ Bd_2 mg/mL
Tomás -Catalá et al. (2017) [74]	MTA-repair HP Angelus (MTA-HP; Angelus) NeoMTA-Plus (N-MTA-P; Avalon Biomed Inc) White MTA Angelus (W-MTA; Angelus)	Indirect and direct contact (SET materials) (ISO 10993-5)	Cell morphology	SEM-EDX (direct contact, 72 h)	Cells attached and merged in all three materials, more cell monolayer structures were evident on the surface of W-MTA. EDX revealed MTA-HP ≈ N-MTA-P ≈ W-MTA in %weight of Ca, C and O.
			Cell Viability (24, 48 & 72 h)	MTT assay	24 h all dilutions: MTA-HP ≈ N-MTA-P ≈ W-MTA ≈ control 48 h undiluted extract: MTA-HP ≈ W-MTA > control 48 h 1:2 dilution: MTA-HP ≈ N-MTA-P ≈ W-MTA ≈ control 48 h 1:4 dilution: W-MTA > control ≈ MTA-HP > N-MTA-P 72 h undiluted extract: W-MTA > N-MTA-P > MTA-HP > control 72 h 1:2 dilution: MTA-HP ≈ N-MTA-P ≈ W-MTA ≈ control 72 h 1:4 dilution: MTA-HP < control ≈ N-MTA-P ≈ W-MTA
			Cell migration (24 & 48 h)	Wound healing–scratch assay	N-MTA-P < control for all dilutions and time points MTA-HP-A > control at 24 h_1:1/1:2 dilutions but < control at 48 h W-MTA-A > control at 24 h_all dilutions but < control at 48 h
Collado-González M et al. (2018) [75]	GI Equia Forte (EF; GC) GI Ionostar Molar (IoM;Voco)	Indirect and direct contact (SET materials) (ISO 10993-5)	Cell morphology (indirect contact, 24 h)	Confocal microscopy (cytoskeletal F-actin)	1:1 extracts EF ≈ control (an organized and stretched stress fiber) 1:1 extracts IoM < control (cell numbers and stretched stress fiber)
			Cell morphology (direct contact, 72 h)	SEM	EF > IoM (cell attachment, morphology and growth)
			Cell Viability (at 24, 48 & 72 h)	MTT assay	24 h all concentrations: Control > EF ≈ IoM 48 h 1:1 dilution: Control ≈ IoM > EF; 48 h 1:2 dilution: IoM ≈ EF ≈ control 48 h 1:4 dilution: IoM ≈ EF ≈ control 72 h 1:1 dilution: EF ≈ control > IoM 72 h 1:2 dilution: control > IoM ≈ EF 72 h 1:4 dilution: control > EF > IoM
			Cell migration (24 and 48 h)	Scratch assay	Control > EF > IoM for all concentrations and study periods
			Cell differentiation	Flow cytometry– Annexin V/7-AAD staining	IoM and EF ≈ control (the percentage of positive expression of mesenchymal markers)
Pedano MS et al. (2018) [76]	Exp-PPL (PPL) Biodentine (Bd; Septodont) Nex-Cem MTA (Nex-MTA; GC) Zinc-oxide eugenol Alganol (ZnO; Kemdent)	Indirect contact (FRESH materials)	Cell viability (24 h)	XTT assay	10% eluates: Bd > PPL ≈ Nex-MTA > ZnO 25% eluates: PPL > Nex-MTA > Bd > ZnO 50% eluates: PPL ≈ Nex-MTA > Bd > ZnO 100% eluates: Nex-MTA > PPL > Bd > ZnO
			Cell proliferation (1, 4 & 7 d)	XTT assay	10% eluates 7d: PPL ≈ Bd ≈ control > Nex-MTA > ZnO 25% eluates 7d: control > Bd > PPL > Nex-MTA > ZnO 50% eluates 7d: control > Bd > PPL > Nex-MTA > ZnO 100% eluates 7d: control > PPL ≈ Bd ≈ Nex-MTA > ZnO
			Cell migration (24 h)	Scratch-wound healing assay	10% and 25% eluates: control ≈ PPL ≈ Nex-MTA > Bd 50% eluates: control ≈Nex-MTA ≈ PPL > Bd 100% eluates: control > PPL > Nex-MTA > Bd
			Cell differentiation (4, 10 & 14 d)	RT-PCR (ALP, OCN, DSPP)	ALP 4 d: differentiation medium > PPL ≈ Bd ≈ Nex-MTA 10 d: differentiation medium ≈ PPL ≈ Bd ≈ Nex-MTA 14 d: differentiation medium > PPL > Bd ≈ Nex-MTA OCN 14d: PPL ≈ Bd > Nex-MTA ≈ differentiation medium DSPP 10 d: PPL ≈ Bd ≈ Nex-MTA ≈ differentiation medium 14 d: Bd > PPL > differentiation medium > Nex-MTA
Tomás-Catalá CJ et al. (2018) [77]	Biodentine (Bd; Septodont)MTA Repair HP Angelus (MTA-HP-A; Angelus) NeoMTA Plus (N-MTA-P; Avalon Biomed Inc)	Indirect and direct contact (SET materials) (ISO 10993-5)	Cell attachment	SEM-EDX (direct contact, 72 h)	SEM showed Bd revealed more cells and with better morphology than MTA-HP-A and N-MTA-P. The EDX revealed that Bd, MTA-HP-A and N-MTA-P had similar percentages of Ca, C and O.
			Cell viability	MTT assay (24, 48 & 72 h)	Undiluted extract: Bd > MTA-HP-A > N-MTA-P > control at 48 h and 72 h 1:2 dilution: Bd > MTA-HP-A ≈ N-MTA-P ≈ control (*p* < 0.01) at 48 h and 72 h 1:4 dilution: Bd > N-MTA-P ≈ control > MTA-HP-A at 72 h
			Cell migration	Scratch assay (at 24 & 48 h)	24 h: Bd > MTA-HP-A ≈ N-MTA-P ≈ control (*p* < 0.01) 48 h: Bd > control for all dilutions; control > N-MTA-P > MTA-HP-A
Lee S-M et al. (2019) [78]	Smart Dentin Replacement (SDR; Dentsply-Sirona) Venus Bulk-fill (VBF; Hereaus Kulzer) Beautifil Bulk flowable (BBF; Shofu) Filtek Z350 XT Flowable (ZFF; 3M)	Indirect contact (Set materials) (ISO 10993-5)	Cell viability	WST assay (24 h) Live/dead Assay (direct visualization with confocal microscopy)	2-mm-cured composite: ≈ 100% cell-viability except for BFF (49%) 4-mm-cured composite: SDR not cytotoxic at all dilutions. VBF & BBF statistically different values (71.05% and 64.43%, respectively) of cell viability at 100% concentration compared to control (*p* < 0.05) but no statistically different cell viability compared to control at 25% and 12.5% concentrations, respectively (~100%, *p* > 0.05) 6-mm-cured composite: SDR and BBF were ~69% and ~6% at 100% concentration (*p* < 0.05), and these resins did not show statistically different cell viability compared to control at 25% and 12.5% (~100%, *p* > 0.05), respectively. In contrast, VBF and ZFF did not reach non-cytotoxic levels (~100%) even at 12.5% dilution.
					At 100% concentrations of SDR, VBF, and ZFF, 6-mm cured composite showed 5~60% live cell numbers compared to the 2-mm cured group. Another bulk-fill resin, BBF, had 5~35% live cells with some dead cells in all groups. At 12.5%, there were full of live cells at all groups while the 4-mm cured ZFF and the 6-mm cured VBF and ZFF revealed fewer live cells (~75%) than the control.
			Cell differentiation (7 days)	ALP staining	6-mm-cured bulk-fill resins showed significantly lower ALP staining than the differentiation media control (*p* < 0.05), while all 2-mm and 4-mm cured bulk-fill resins showed similar ALP staining, except for 4-mm-cured BBF. ALP staining from the bulk-fill resins was ranked as follows: 2-mm > 4-mm > 6-mm cured. The flowable resin, ZFF, exhibited the least amount of ALP staining between the experimental groups.
López-García et al. (2019) [79]	Activa *Kids* (Activa; Pulpdent) GI Ionolux (Voco) Riva Light Cure (Riva; SDI)	Indirect and direct contact (Set materials) (ISO 10993-5)	Cell morphology (indirect contact)	Immunofluorescence	Activa > cell density and spreading than Riva > Inolux
			Cell attachment/adhesion (direct contact)	SEM	Activa showed well-adhered fibroblastic cells with multiple cytoplasmic extensions. Riva showed less density and fewer cells than Activa. Ionolux induced drastic reduction in cell density and attachement.
			Cell viability	MTT assay (1, 2 & 4 days)	24 h - Undiluted extracts: Activa ≈ control > Riva > Ionolux (*p* < 0.01) 24 h–1:2 dilution: Activa ≈ control ≈ Riva > Ionolux 24 h–1:4 dilution: Activa ≈ control > Riva > Ionolux 48 h-Undiluted extracts: Ionolux < Activa & Riva (*p* < 0.01) < control (*p* < 0.01) 48 h–1:2 dilution: Activa & Riva & Ionolux ≈ control 48 h–1:4 dilution: Activa & Riva & Ionolux ≈ control
					72 h-Undiluted extracts: Control > Activa > Riva > Ionolux 72 h–1:2 dilution: Control > Activa > Riva > Ionolux 72 h–1:4 dilution: Activa ≈ control; Riva & Ionolux < control
			Cell migration	Wound healing assay	Activa ≈ control at all dilutions except 1:2 at 72 h Riva < migration than control except 1:4 dilution Ionolux < migration than control except 1:4 dilution at 24 h and 48 h
Dou L et al. (2020) [80]	Dycal (Dentsply-Sirona) Pro-Root MTA (MTA; Dentsply-Sirona) iRoot BP (iRoot; Innovative Bioceramix) Platelet-rich Fibrin (PRF) Concentrated Growth Factors (CGF)	Indirect contact (Set materials)	Cell viability	Trypan Blue Staining (1, 3 & 7 days) Flow cytometry– Annexin V-PI (1, 3 & 7 days) Cell Cycle(1, 3 & 7 days)	Dycal < cell viability than MTA ≈ iRoot ≈ PRF ≈ CGF ≈ control at 1, 3 & 7 days
					Dycal > apoptotic cells than MTA ≈ iRoot ≈ CGF ≈ control at 1, 3 & 7 days Days 1 & 3: no significant differences among the groups Day 7: CGF showed less cells in G_0_/G_1_-phase compared to MTA & Dycal
			Cell proliferation	CCK-8	Day 1: Dycal < cell proliferation than all groups; MTA ≈ iRoot ≈ PRF ≈ CGF ≈ control. Day 3: PRF & CGF > cell proliferation than control & MTA, but ≈ iRoot; Dycal < all groups Day 7: CGF > cell proliferation than iRoot & MTA, but ≈ control &PRF; Dycal < all groups
			Cell differentiation (1,3 & 7 days)	ALP staining	Days 1 & 3: MTA > ALP-activity than control; Control ≈ iRoot ≈ PRF ≈ CGF ≈ Dycal Day 7: Dycal < ALP-activity than CGF; CGF ≈ control ≈ MTA ≈ iRoot ≈ PRF

* Direct contact was considered when the cells were seeded on top of the materials. When the material was placed on a transwell insert or materials’ eluates were used, it was considered INDIRECT contact.

**Table 3 materials-13-02670-t003:** Included in vivo studies (immature permanent teeth).

**Authors**	**Study Type**	**Hemostasia**	**Materials Used**	**Etched Pulp?**	**Evaluation Period(s)**	**Bridge Formation**	**Inflammation**	**Sample**
Hebling J et al. (1999) [81]	CCT	Sterile cotton pellets + sterile saline	Calcium-hydroxide saline paste (CH) + calcium-hydroxide cement (Dycal; Dentsply-Sirona) All Bond 2 (AB2; Bisco)	No (CH), Yes (AB2)	7 days 30 days 60 days	CH: 3/4 teeth showed complete bridge formation at 30 days. 4/4 teeth showed complete bridge below exposed area at 60 days. AB2: 0/6 teeth showed completed bridge at 30 or 60 days (0/12 in total). All of them showed modest bridge formation at 60 days.	CH: 1/4 teeth showed moderate and 3/4 slight inflammation at day 7. At day 30, 4/4 teeth showed slight inflammatory reaction. No tooth showed severe inflammation AB2: 1/6 teeth showed severe inflammatory reaction, 3/6 moderate and 2/6 slight inflammation at day 7. At day 30, 3/6 showed moderate and 3/6 slight inflammatory reaction.	32 premolars (12–15 years old patients)
Do Nascimento AB et al. (2000) [82]	CCT	Sterile paper cones + sterile saline	Calcium-hydroxide saline paste (CH; Pathfinder associates) + calcium hydroxide cement (Dycal; Dentsply-Sirona)Resin-modified glass-ionomer cement (Vit; Vitrebond; 3M Oral Care)	No	5 days 30 days 120+ days	CH: 1/4 teeth showed complete bridge at 30 days. 5/5 teeth showed complete bridge at 120+ days. Vit: 0/6 teeth showed complete bridge at 30 days. 0/5 teeth showed complete bridge at 120+ days.	CH: at day 5, 0/6 teeth showed no inflammation, 4/6 teeth showed slight, 2/6 moderate and 0/6 severe inflammation. At day 30, 3/4 showed slight and 1/4 moderate inflammation. Vit: 0/5 teeth showed none inflammatory reaction, 1/5 teeth showed slight and 4/5 moderate inflammation at day 5. At day 30, 2/6 showed slight and 4/6 moderate inflammatory reaction.	34 premolars (11–17 years old patients)
Costa CAS et al. (2001) [83]	CCT	Sterile paper cones + sterile saline	Calcium-hydroxide saline paste (CH; Pathfinder associates) + calcium hydroxide cement (Dycal; Dentsply-Sirona) Clearfil Liner Bond 2 (CLB2; Kuraray Noritake)	No	5 days 30 days 120+ days	CH: 1/4 teeth showed complete bridge at 30 days. 5/5 teeth showed complete bridge at 120+ days. CLB2: 0/5 teeth showed complete bridge at 30 days. 0/6 teeth showed complete bridge at 120+ days.	CH: at day 5, 1/6 teeth showed none inflammatory reaction, 3/6 teeth showed slight, 2/6 moderate and 0/6 severe inflammation. At day 30, 3/4 showed slight and 1/4 moderate inflammation. CLB2: 4/6 teeth showed slight and 2/6 moderate inflammation at day 5. At day 30, 2/5 showed slight and 3/5 moderate inflammatory reaction.	36 premolars (11–17 years old patients)
Ersin EK et al. (2005) [84]	CCT	3% H_2_O_2_	Calcium-hydroxide saline paste (CH) + calcium hydroxide cement (Dycal; Dentsply-Sirona) Prime&Bond 2.1 (P&B2.1; Dentsply-Sirona)	No	7 days 90 days	CH: 5/5 teeth showed complete bridge after 90 days. P&B2.1: 0/5 teeth showed complete bridge after 90 days.	CH: 5/5 teeth showed slight inflammatory reaction at day 7. P&B2.1: at 7 days, 5/5 teeth showed moderate acute inflammatory reaction. 0/5 teeth showed severe inflammation or necrotic tissue.	20 premolars (mean age 12.6 years) Age range not available
Olsson H et al. (2005) [85]	RCT	Continuous irrigation with sterile saline	Calcium-hydroxide saline paste (CH) Enamel matrix derivative (EMD; Emdogain, BIORA)	No	12 weeks	CH: 9/9 teeth showed complete bridge at 12 weeks. EMD: 0/9 teeth showed complete bridge at 12 weeks.	12 weeks	18 premolars (12–16 years old patients)
Silva GAB et al. (2006) [86]	RCT	Sterile cotton pellets + sterile saline	Calcium-hydroxide powder (CH; PA Biodinamica) + calcium hydroxide cement (Dycal; Dentsply-Sirona) Single Bond (SB; 3M Oral Care)	No (CH), Yes (10%-37%; SB)	1 day 3 days 7 days 30 days	CH: 5/5 teeth showed complete bridge at day 30. SB-10% Etch: 0/5 teeth showed complete bridge at day 30. SB-37% Etch: 0/5 teeth showed complete bridge at day 30.	CH: 0/20 showed severe inflammatory reaction at days 1-3-7 or 30 (5/time period). 10/20 teeth showed slight and 10/20 teeth showed moderate inflammation at 1-3-7 or 30 days. SB-10% Etch: 0/20 showed severe inflammatory reaction at days 1-3-7 or 30 (5/time period). 1/20 teeth showed none/few inflammatory cells at d1. 5/20 teeth showed slight and 14/20 moderate inflammatory reaction. SB-37% Etch: 0/20 showed severe inflammatory reaction at days 1-3-7 or 30 (5/time period). 1/20 teeth showed none/few inflammatory cells at d1. 10/20 teeth showed slight and 9/20 moderate inflammation at 1-3-7 or 30d.	81 premolars (12–17 years old patients)
Sawicki L et al. (2008) [87]	RCT	Sterile cotton pellets + sterile saline	Calcium-hydroxide cement (Life; Kerr) White Pro-Root MTA (W-MTA; Dentsply-Sirona)	No	47+ days	Life: 11/14 teeth showed complete bridge after 47+ days. WMTA: 28/30 teeth complete bridge. 4 teeth lost (not reported)	47+ days!	48 premolars (10–18 years old patients)
Azimi S et al. (2014) [88]	RCT	Sterile cotton pellets + sterile saline	White Pro-Root MTA (W-MTA; Dentsply-Sirona) iRoot BP (iRoot; Innovative Bioceramix)	No	6 weeks	WMTA: 8/12 teeth showed complete bridge formation at 6 weeks. iRoot: 7/12 teeth showed complete bridge at 6 weeks.	6 weeks!!	24 premolars (12–16 years old)
Swarup SJ et al. (2014) [89]	RCT	Sterile cotton pellets + sterile saline	Calcium-hydroxide cement (Dycal; Dentsply) MTA Angelus (MTA-A; Angelus) Nanohydroxyapatite (Hap; Orthogran)	No	15 days 30 days	Dycal: 2/5 had a complete bridge at 30d. MTA-A: 4/5 had complete bridge 30d. Hap: 4/5 teeth had complete bridge.	Dycal: 4/5 showed moderate and 1/5 severe inflammatory reaction day 15. 1/5 teeth showed moderate and 4/5 slight inflammation at day 30. MTA-A: 3/5 teeth showed none and 2/5 teeth showed slight inflammatory reaction at day 15. At day 30, 4/5 teeth showed none or few inflammations and 1/5 showed slight inflammatory reaction. Hap: 3/5 showed moderate inflammatory reaction and 2/5 severe inflammation (day 15). 3/5 showed no inflammation and 2/5 slight inflammatory reaction at day 30.	30 premolars (11–15 years old)
Banava S et al. (2015) [90]	RCT	Sterile cotton pellets + sterile saline	Calcium-hydroxide cement (Dycal; Dentsply-Sirona) White-MTA (WMTA; Dentsply-Sirona)Pulpdent Multi-Cal Liner (PML; Pulpdent)	No	6 weeks	Dycal: 7/8 teeth showed bridge at 6w. WMTA: 12/16 teeth sowed bridge formation at 6w. PML: 2/8 teeth showed bridge formation at 6w.	6 weeks!!	32 premolars (13–20 years old)

**Table 4 materials-13-02670-t004:** Included in vivo studies (mature permanent teeth).

Authors	Study Type	Hemostasia	Materials Used	Etched Pulp?	Evaluation Period(s)	Bridge Formation	Inflammation	Sample
Demarco FF et al. (2001) [91]	RCT	Sterile cotton pellets + sterile saline	Calcium-hydroxide powder (CH; Labrynth Produtos) + calcium-hydroxide cement Hydro C (Dentsply-Sirona) Scotchbond Multipurpose (SBMP; 3M) Clearfil Liner Bond 2 (CLB2; Kuraray Noritake)	Yes (SBMP), No (CH and CLB2)	30 days 90 days	CH: 2/2 complete bridges formed at 30 and 2/2 at 90 days SBMP: 0/4 teeth with bridges formed at 30 and 0/4 at 90 days. CLB2: 1/4 teeth showed bridge at 30 days. 2/4 teeth showed bridge formation at 90 days.	CH: 0/2 teeth showed severe inflammation or necrosis at 30 days. SBMP: 1/4 teeth showed severe inflammation or necrosis at 30 days. CLB2: 0/4 teeth showed severe inflammation or necrosis at 30 days.	20 molars (20–27 years old patients)
Accorinte MLR et al. (2005) [92]	RCT	Sterile cotton pellets + sterile saline	Calcium-hydroxide powder (CH; Labrynth Produtos) + calcium-hydroxide cement (Dycal; Dentsply-Sirona) Scotchbond Multipurpose (SBMP; 3M Oral Care)	Yes (SBMP), No (CH)	60 days	CH: 100% (5/5) of the teeth showed brige formation at day 60. SBMP: 0% (0/5) of the teeth showed bridge formation	60 days	25 premolars (15–25 years old patients)
Accorinte MLR et al. (2006) [93]	RCT	Sterile cotton pellets + sterile saline	Calcium-hydroxide powder (CH; Labrynth Produtos) + calcium-hydroxide cement (Dycal; Dentsply-Sirona) Scotch Bond Multipurpose (SBMP; 3M Oral Care)	Yes (SBMP), No (CH)	30 days 60 days	CH: 100% (5/5) of the teeth showed brige formation at 30 and 60 days. SBMP: 0% (0/5) of the teeth showed bridge formation at 30 or 60 days.	CH: no to mild inflammatory reaction all teeth (5/5) at 30 days. SBMP: 2/5 teeth (40%) showed pulp necrosis at day 30.	40 premolars (15–25 years old patients)
Iwamoto CE et al. (2006) [94]	RCT	Sterile cotton pellets + sterile saline	Calcium-hydroxide cement (Dycal; Dentsply-Sirona) White Pro-Root MTA (W-MTA; Dentsply-Sirona)	No	112+ days	Dycal: 18/23 teeth developed bridge formation at 110+ days. W-MTA: 20/22 teeth developed bridge formation.	112+ days	48 molars (18–60 years old patients)
Accorinte MLR et al. (2007) [95]	CCT	Sterile cotton pellets + sterile saline; sterile cotton pellets + 2.5% sodium hypochlorite	Calcium-hydroxide cement (Life; Kerr)	No	30 days 60 days	Life: 2/10 teeth (20%) showed bridge formation at 30 days. At day 60, 6/10 teeth showed complete bridge formation. At day 60, only 1/10 teeth showed absent bridge.	Life: At day 30, 8/10 teeth showed no inflammation and 1/10 showed mild inflammatory reaction. Only 1/10 teeth showed severe inflammation.	40 premolars (15–30 years old patients)
Elias RV et al. (2007) [96]	RCT	Sterile cotton pellets + 2.5% sodium hypochlorite	Calcium-hydroxide powder (CH; Labrynth Produtos) + calcium-hydroxide cement Hydro C (Dentsply-Sirona) Clearfil SE Bond (CSE; Kuraray Noritake)	No	30 day 90 days	CH: 5/5 complete bridge after 30 days and 5/5 at 90 days. CSE: 1/8 specimens showed dentin deposition at the interface (complete bridge) at 90 days. 0/8 teeth showed complete bridge at 30 days. 3/8 specimens showed no dentin deposition at all.	CH: 5/5 teeth showed none or slight inflammation after 30 days for all specimens. CSE: 4/8 specimens showed no inflammation; 3/8 specimens slight inflamm; 1/8 severe inflammation at 30 days.	26 molars (average 25 years) Age range not provided
Accorinte MLR et al. (2008) [53]	CCT	Sterile cotton pellets + sterile saline	Calcium-hydroxide cement (Life; Kerr) Pro-Root MTA (MTA; Dentsply-Sirona)	No	30 days 60 days	Life: 2/10 teeth showed complete bridge at 30 days and 6/10 complete bridge after 60 days MTA: 3/10 teeth showed complete bridge at 30 days and 5/10 teeth complete bridge after 60 days	Life: all teeth showed absent (8/10) or mild (2/10) inflammatory reaction at 30 days. 1/10 teeth showed slight and 1/10 moderate inflammatory reaction. No teeth showed necrosis or abscess formation at 30 days.MTA: all teeth showed absent (9/10) or mild (1/10) inflammatory reaction at 30 days. No teeth showed necrosis or abscess formation at 30 days.	40 premolars (15–30 years old patients)
Accorinte MLR et al. (2008) [52]	CCT	Sterile cotton pellets + sterile saline	Calcium-hydroxide powder (CH) + calcium-hydroxide cement (Life; Kerr) MTA Angelus (MTA-A; Angelus)	No	30 days 60 days	CH: 6/10 teeth showed complete bridge at 30 days and 8/10 teeth showed complete bridge after 60 days MTA-A: 4/10 teeth showed complete bridge at 30 days and 7/10 complete bridge after 60 days	CH: 8/10 teeth showed absent or slight inflammatory reaction at 30 days. No teeth showed necrosis or abscess formation at 30 days. MTA-A: 8/10 teeth showed absent or slight inflammatory reaction at 30 days. 1/10 teeth showed severe inflammation (abscess formation) at 30 days.	40 premolars (15–30 years old patients)
Accorinte MLR et al. (2008) [97]	CCT	Sterile cotton pellets + sterile saline	Calcium-hydroxide cement (Dycal; Dentsply-Sirona) Clearfil Liner Bond 2V (C2V; Kuraray Noritake) Clearfil SE Bond (CSE; Kuraray Noritake)	No	30 days 90 days	Dycal: 8/9 teeth with complete bridge formed at day 90. No CH treated teeth extracted at 30 days. C2V: 1/4 teeth complete bridge at 30 days. 1/6 teeth with complete bridge at day 90 and 2/6 with absence or discrete bridge CSE: 0/5 teeth complete bridge at 30 days. 1/6 teeth with complete bridge at day 90 and 4/6 with absence or discrete bridge	Dycal: no teeth were extracted at 30 days for CH group. C2V: All teeth had slight (2/6) or moderate (3/6) inflammation after 30 days including 1/6 teeth with abscess. CSE: All teeth had slight (3/6) or moderate (2/6) inflammation after 30 days including 1/6 teeth with abscess.	34 premolars (15–30 years old patients)
Lu Y et al. (2008) [98]	RCT	Sterile cotton pellets + 2% chlorhexidine + sterile saline	Calcium-hydroxide cement (Dycal; Dentsply-Sirona) Clearfil SE Bond (CSE; Kuraray Noritake)	No	7 days 30 days 90 days	Dycal: 1/6 teeth complete bridge at day 30. 5/7 teeth complete bridge deposits after 90 days. CSE: 0/7 teeth complete bridge after 30 and 0/7 at 90 days.	Dycal: 2/7 teeth showed slight, 4/7 moderate and 1/7 teeth showed severe inflammatory reaction at day 7. At day 30, 5/6 teeth showed slight and 1/6 moderate inflammatory reaction. CSE: 6/7 teeth showed slight and 1/7 moderate inflammatory reaction at both 7 and 30 days.	45 molars (20–25 years old patients)
Min K-S et al. (2008) [99]	RCT	Sterile cotton pellets + sterile saline	Calcium-hydroxide cement (Dycal; Dentsply-Sirona) Pro-Root MTA (MTA; Dentsply-Sirona)	No	2 months	Dycal: 6/10 teeth complete bridge. 4/10 total absence of bridge. MTA: 100% complete bridge formation. Bridges were significantly thicker with MTA	2 months!!	20 molars (2–50 years old patients)
Nair PNR et al. (2008) [100]	RCT	1% sodium hypochlorite + sterile saline + paper points	Calcium-hydroxide cement (Dycal; Dentsply-Sirona) Pro-Root MTA (MTA; Dentsply-Sirona)	No	7 days 30 days 90 days	Dycal: 1/5 teeth complete bridge and 3/5 partial bridge at day 30. At day 90, 2/4 teeth partial bridge and 0/4 complete bridge at 90 days. Thick bridges with tunnel defects. MTA: 3/6 teeth with complete bridge formation at day 30. 4/5 teeth complete bridge formation at day 90. MTA showed thicker and less variable bridges than Dycal.	Dycal: at day 7, 2/4 teeth showed inflammatory reaction, 1 of them with severe inflammation. At day 30, 1/5 teeth showed severe inflammatory reaction. MTA: 5/6 teeth showed absence of inflammatory reaction and 1/6 showed slight inflammation at day 7. At day 30, 6/6 samples showed no inflammatory reaction.	33 molars (18–30 years old patients)
Accorinte MLR et al. (2009) [101]	RCT	Sterile cotton pellets + sterile saline	Pro-Root MTA (MTA; Dentsply-Sirona) MTA Angelus (MTA-A; Angelus)	No	30 days 60 days	MTA: 5/8 teeth had complete bridge at day 30. 5/9 teeth showed complete bridge at day 60.MTA-A: 5/8 teeth had complete bridge at day 30. 6/10 teeth showed complete bridge at day 60.	MTA: 2/8 teeth showed no inflammation and 6/8 slight inflammatory reaction at day 30. MTA-A: 3/8 teeth showed no inflammation at day 30. 4/8 showed mild inflammatory reaction and 1 tooth showed abscess at day 30.	35 premolars (25–42 years old patients)
Kiatwateeratana T et al. (2009) [102]	RCT	Moistened sterile cotton pellets	Calcium-hydroxide powder (CH) Enamel matrix derivative (EMD; Emdogain, BIORA)	No	6 months	CH: 10/13 teeth showed complete bridge formation EMD: no tooth showed bridge formation	6 months!	26 premolars (13–22 years old)
Parolia A et al. (2010) [103]	CCT	Moistened sterile cotton pellets	Calcium-hydroxide cement (Dycal; Dentsply-Sirona) Pro-Root MTA (MTA; Dentsply-Sirona)Propolis powder (Propolis; Ecuadorian Rainforest LLC) mixed with 70% ethanol	No	15 days 45 days	Dycal: 5/6 teeth showed bridge formation at 45 days. MTA: 6/6 teeth showed bridge formation at day 45. Propolis: 6/6 teeth showed bridge formation at 45 days.	Dycal: 6/6 teeth showed none or mild inflammation at day 15. MTA: all teeth (6/6) showed none or mild inflammation at day 15. Propolis: 6/6 teeth showed none or mild inflammation at day 15.	36 premolars (15–25 years old)
Nowicka A et al. (2013) [104]	CCT	Sterile cotton pellets + sterile saline	White Pro-Root MTA (W-MTA; Dentsply-Sirona) Biodentine (Biodentine; Septodont)	No	6 weeks	W-MTA: 7/11 teeth had complete bridge. All teeth formed bridges. Biodentine: 6/11 teeth had complete bridge. All teeth formed bridges.	6 weeks!	28 molars (19–28 years old patients)
Nowicka A et al. (2016) [105]	RCT	Sterile cotton pellets + sterile saline	Calcium hydroxide paste (Calcipast; Cerkamed) + Calcium hydroxide cement (Life, Kerr) Single Bond Universal (SBU; 3M Oral Care)	No	6 weeks	Calcipast: all teeth showed bridge formation with 5/11 teeth showing complete bridge at 6 wk SBU: none of the teeth showed complete bridge formation. 7/11 teeth showed no bridge formation at 6 wks.	6 weeks!	28 molars (19–28 years old patients)
Bakhtiar H et al. (2017) [106]	RCT	Sterile cotton pellets	Theracal LC (Theracal; Bisco) Biodentine (Biodentine; Septodont) Pro-Root MTA (W-MTA; Dentsply-Sirona)	No	8 weeks	Theracal: 2/9 teeth showed no bridge formation at 8 wks. 1/9 teeth showed complete bridge at 8 wks. Biodentine: All teeth (9/9) showed a complete bridge formation at 8 wks. W-MTA: 5/9 teeth showed a complete bridge at 8 wks. 4/9 teeth showed an incomplete bridge at 8 wks.	8 weeks!	27 molars (18–32 years old patients)
Jalan AL et al. (2017) [107]	RCT	Sterile cotton pellets + sterile saline	Calcium-hydroxide cement (Dycal; Dentsply-Sirona) Biodentine (Biodentine; Septodont)	No	45 days	Dycal: 1/20 teeth showed no bridge formation. 4/20 teeth showed a complete bridge. Biodentine: all teeth showed bridge formation. 16/20 teeth complete bridge.	45 days	40 premolars (15–25 years old)
Mehrvarzfar P et al. (2018) [108]	RCT	Sterile cotton pellets + sterile saline	White Pro-Root MTA (W-MTA; Dentsply-Sirona) W-MTA + Treated Dentin Matrix (W-MTA/TDM; Dentsply-Sirona)	No	6 weeks	W-MTA: 11/11 complete bridge was present in all specimens at 6 wk. W-MTA/TDM: 11/11 complete bridge was present in all specimens at 6 wks.	6 weeks!	26 molars (15–31 years old patients)

**Table 5 materials-13-02670-t005:** Relative risks (95% CIs) and certainty in the evidence for pulp inflammation at day 30 after direct pulp-capping treatment on human pulp tissue.

Comparison	Direct Comparison		Indirect Comparison	
	Relative Risk (95% CIs)	Evidence Level	Relative Risk (95% CIs)	Evidence Level
Pro-Root MTA vs. calcium-hydroxide (CH) powder	No studies available	Not available	No studies available ^1^	Not available
Pro-Root MTA vs. CH cements (Dycal, Life, Calcipast, …)	0.29 [0.01, 5.79]	VERY LOW ^2 b^	0.39 [0.04, 3.47]	LOW ^a^
MTA-like cements (Pro-Root MTA, Angelus MTA, …) vs. CH powder	3.00 [0.14, 65.90] ^3^	VERY LOW ^3 b^	2.67 [0.29, 24.24]	LOW ^a^
MTA-like cements vs. CH cements	0.29 [0.01, 5.79]	VERY LOW ^b^	0.87 [0.16, 4.71]	LOW ^a^
MTA-like cements vs. ALL CH-based materials (CH cements + powder)	0.93 [0.15, 5.79]	LOW	1.17 [0.26, 5.26]	LOW ^a^
CH cements vs. CH powder	No studies available	Not available	5.41 [0.27, 108.93]	LOW ^a^
Calcium-silicate cements (Biodentine, iRoot, …) vs. MTA-like cements	No studies available ^4^	Not available	No studies available ^4^	Not available
Resin-based vs. resin-free materials	2.41 [0.48, 12.03]	VERY LOW ^b^	1.33 [0.44, 4.04]	MODERATE ^a^

^1^ 11 studies available for indirect comparison between Pro-Root MTA (3/11 studies) vs. CH powder (8/11 studies). However, all of them retrieved 0 events (no severe inflammation or necrosis/abscess formation reported). Therefore, quantitative analysis is not possible. ^2^ Only 2 studies available for direct comparison between Pro-Root MTA vs. CH cements [53,100]. ^3^ Only 1 study available for direct comparison between MTA-like cements vs. CH powder [52]. ^4^ No studies available evaluating inflammation of calcium-silicate cements (Biodentine, iRoot, …) at 30 days or before. ^a^ Level of the evidence downgraded one level due to high risk of bias in all studies and due to imprecision as very few studies are available with very few events. ^b^ Level of the evidence very low due to high risk of bias in all studies and due to imprecision as very few studies are available with very few events.

**Table 6 materials-13-02670-t006:** Relative risks (95% CIs) and certainty in the evidence for pulp inflammation at day 15 after direct pulp-capping treatment on human pulp tissue.

Comparison	Direct Comparison		Indirect Comparison	
	Relative Risk (95% CIs)	Evidence Level	Relative Risk (95% CIs)	Evidence Level
Pro-Root MTA vs. calcium-hydroxide (CH) powder	No studies available ^1^	Not available ^1^	No studies available ^1^	Not available ^1^
Pro-Root MTA vs. CH cements (Dycal, Life, Calcipast, …)	Not available ^2^	Not available	0.54 [0.02, 15.30]	VERY LOW ^a^
MTA-like cements (Pro-Root MTA, Angelus MTA, …) vs. CH powder	No studies available ^3^	Not available	No studies available ^3^	Not available ^3^
MTA-like cements vs. CH cements	0.33 [0.02, 6.65] ^3^	VERY LOW ^a^	Not available ^3^	VERY LOW ^a^
MTA-like cements vs. ALL CH-based materials (CH cements + powder)	0.33 [0.02, 6.65] ^3^	VERY LOW ^a^	Not available ^3^	VERY LOW ^a^
CH cements vs. CH powder	No studies available	Not available	No studies available ^4^	Not available
Calcium-silicate cements (Biodentine, iRoot, ...) vs. MTA-like cements	No studies available ^5^	Not available	No studies available ^5^	Not available
Resin-based vs. resin-free materials	No studies available ^6^	Not available	No studies available ^6^	Not available

^1^ Only one study available for Pro-Root MTA (none for CH powder) [103]. ^2^ One study for direct comparison between Pro-Root MTA and CH cements (Life, Dycal, Calcipast, etc.) at day 15. However, it gave 0 events (no severe inflammation or necrosis/abscess formation reported). Therefore, quantitative analysis was not possible. ^3^ Only two studies available for MTA-like cements (both of them direct comparison against CH cements, none for CH powder) [89,103]. ^4^ Only two studies available, both for CH cements [89,103]. ^5^ No studies available evaluating inflammation of calcium-silicate cements (Biodentine, iRoot, etc.) at 30 days or before. ^6^ No studies available evaluating inflammation of resin-based materials at 15 days. ^a^ Level of the evidence very low due to high risk of bias in all studies and due to imprecision as very few studies are available with very few events.

**Table 7 materials-13-02670-t007:** Relative risks (95% CIs) and certainty in the evidence for pulp inflammation up to day 7 after direct pulp-capping treatment on human pulp tissue.

Comparison	Direct Comparison		Indirect Comparison	
	Relative Risk (95% CIs)	Evidence Level	Relative Risk (95% CIs)	Evidence Level
Pro-Root MTA vs. calcium hydroxide powder	No studies available	Not available	Not available ^1^	Not available ^1^
Pro-Root MTA vs. CH cements (Dycal, Life, Calcipast, …)	0.24 [0.01, 4.72] ^2^	VERY LOW	0.34 [0.02, 6.17]	VERY LOW ^b^
MTA-like cements (Pro-Root MTA, Angelus MTA, …) vs. CH powder	No studies available	Not available	Not available ^3^	Not available ^3^
MTA-like cements vs. CH cements	0.24 [0.01, 4.72] ^2^	VERY LOW ^2^	0.34 [0.02, 6.17]	VERY LOW ^b^
MTA-like cements vs. ALL CH-based materials (CH cements + powder)	0.24 [0.01, 4.72] ^2^	VERY LOW ^2^	1.37 [0.07, 25.71]	VERY LOW ^2^
CH cements vs. CH powder	No studies available	Not available	15.42 [0.79, 299.22]	LOW ^a^
Calcium-silicate cements (Biodentine, iRoot, ...) vs. MTA-like cements	No studies available ^4^	Not available	No studies available ^4^	Not available
Resin-based vs. resin-free materials	0.84 [0.12, 5.74]	VERY LOW ^b^	0.84 [0.12, 5.74] ^5^	VERY LOW ^b^

^1^ Six studies available for indirect comparison (one Pro-Root MTA and five for CH powder); however, all of them retrieved 0 events (no severe inflammation or necrosis/abscess formation reported). Therefore, quantitative analysis was not possible. ^2^ Only one study for Pro-Root MTA [100]. ^3^ 6 studies available for indirect comparison (1 MTA-like cements and five for CH powder); however, all of them retrieved 0 events (no severe inflammation or necrosis/abscess formation reported). Therefore, quantitative analysis was not possible. ^4^ No studies available evaluating inflammation of calcium-silicate cements (Biodentine, iRoot, ...) at 30 days or before. ^5^ All events (severe inflammation or necrosis/abscess formation) are in studies with direct comparison. No events available for indirect comparison. ^a^ Level of the evidence downgraded one level due to high risk of bias in all studies and due to imprecision as very few studies are available with very few events. ^b^ Level of the evidence very low due to high risk of bias in all studies and due to imprecision as very few studies are available with very few events.

**Table 8 materials-13-02670-t008:** Relative risks (95% CIs) and certainty in the evidence for bridge formation after direct pulp-capping treatment on human pulp tissue.

Comparison	Direct Comparison		Indirect Comparison	
	Relative Risk (95% CIs)	Evidence Level	Relative Risk (95% CIs)	Evidence Level
Pro-Root MTA vs. Calcium-hydroxide (CH) powder	No studies available	Not available	1.64 [0.98, 2.77]	MODERATE ^a^
Pro-Root MTA vs. CH cements (Dycal, Life, Calcipast, …)	0.39 [0.22, 0.67] **	HIGH	0.56 [0.42, 0.74] ***	HIGH
MTA-like cements (Pro-Root MTA, Angelus MTA, …) vs. CH powder	1.50 [0.66, 3.43] ^1^	VERY LOW ^1^	2.02 [1.21, 3.36] **	MODERATE ^a^
MTA-like cements vs. CH cements	0.41 [0.23, 0.73] **	HIGH	0.59 [0.45, 0.78] ***	HIGH
MTA-like cements vs. ALL CH-based materials (CH cements + powder)	0.66 [0.47, 0.92] *	HIGH	0.80 [0.62, 1.03]	HIGH
CH cements vs. CH powder	Not available	Not available	3.23 [2.00, 5.20] ***	MODERATE ^a^
Calcium-silicate cements (Biodentine, iRoot, ...) vs. MTA-like cements	0.84 [0.43, 1.65]	MODERATE ^a^	0.81 [0.47, 1.40]	MODERATE ^a^
Biodentine vs. Pro-Root MTA	0.50 [0.04, 6.43]	VERY LOW ^3 b^	0.86 [0.36, 2.02]	LOW ^b^
Calcium-silicate cements vs. CH powder	Not available	Not available	1.80 [0.95, 3.40]	LOW ^a b^
Calcium-silicate cements vs. CH cements	0.25 [0.10, 0.62] ** ^2^	VERY LOW ^2^	0.47 [0.15, 1.44]	LOW ^a b^
Resin-based vs. resin-free materials	3.69 [2.23, 6.12] ***	HIGH ^a^	Not performed ^4^	Not performed

^1^ Only one study available for direct comparison between MTA-like cements vs. CH powder [52]. ^2^ Only 1 study available for direct comparison between calcium-silicate cements vs. CH cements [107]. ^3^ Only two studies available for direct comparison between Biodentine vs. Pro-Root MTA [104,106]. ^4^ The direct comparison was strong enough (13 studies). ^a^ Level of the evidence downgraded one level due to high risk of bias in all studies. ^b^ Level of the evidence downgraded one level due to high risk of bias in all studies and imprecision. * The result is statistically significant (*p* < 0.05); ** The result is statistically significant ( *p* <0.01); *** The result is statistically significant (*p* < 0.001 or lower).

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
