# Peer review of "Cytotoxicity and Bioactivity of Dental Pulp-Capping Agents towards Human Tooth-Pulp Cells: A Systematic Review of In-Vitro Studies and Meta-Analysis of Randomized and Controlled Clinical Trials"

_materials, 2020, doi:10.3390/ma13122670_

Round 1
Reviewer 1 Report
Manuscript ID: materials-820392
Lines 75-90: The authors write that pro-Root MTA was the first calcium sylicate cement used as pulp capping with several limitations. They afterwards propose that one of the most innovative field of research is the developing of resin-based agents. However the authors (correctly) reports that the main drawback of these cements is the lack of biocompatibility of the resin monomers released which are in contacts with pulp cells. After that they write that “resins are not toxic by definition and researchers are already working to develop biocompatible, naturally derived resin blends that may be suitable for biomedical applications”. This statement must be clarified. First, literature reported by the authors for supporting this information is about studies from a chemical point of view (ref. 29 for instance). In order to affirm that a resin is “biocompatible”, authors should add literature reporting in vitro studies, where resins are tested on cell cultures in order to investigate cytotoxicity and molecules involved in biocompatibility (cell attachment as an example). Second, resins are not toxic for definition when their polymerization is 100% completed in the oral cavity. There is a large amount of literature reporting that residual monomers leach into the surrounding aqueous environment after incomplete polymerization of the organic matrix of resin materials can interfere with adjacent tissues. Are there evidence that the new resins don’t release monomers in the oral cavity after incomplete polymerization? Are there evidence that they don’t induce immunomodulatory reactions involving cells like macrophages in the oral cavity?
Lines 36-43 (Study selection) and section 3.1 (Qualitative analysis of in vitro studies):
When the authors write “In vitro studies were included when human dental pulp (stem) cells of primary origin were (in)directly exposed to pulp-capping agents”, should clarify the meaning of an indirect exposition to these agents. How a agent can be defined cytotoxic or not for pulp cells if it is not directly in contact with cell cultures? Moreover, why “stem” is always reported in brackets? The percentage of the mesenchymal stem cell population in the pulp is very low compared to the percentage of fibroblasts, as an example. If primary cells extracted from the pulp are not sorted by specific CD markers and therefore stem cells are isolated, we don’t know what kind of cells are directly in contact with the materials. I suggest to write only “dental pulp cells”.
Tables
Table 2:
When in the method column is reported “flow cytometry” as a method for evaluating cell viability, the authors should be more precise reporting the assay used (Annexin V-PI?).
Zhang et al. 2015 [55]: What kind of cell apoptosis assay? What kind of cell double-labeling assay?
Chung et al. 2016 [56]: Is it phase-contrast microscopy?
Jeanneu et al. 2017 [24]: sometimes the authors report the MTT assay as a proliferation assay and sometimes as a viability assay. The authors should uniform the classification method of the various assays.
Collado-Gonzalez et al. 2018 [115]: What kind of molecules have been investigated by confocal microscopy in order to visualize the fiber organization?

Author Response
Please, see attached file.

Reviewer 2 Report
The aims of the present exhaustive systematic review “Cytotoxicity and bioactivity of dental pulp-capping agents towards human tooth-pulp cells: a systematic review of in-vitro studies and meta-analysis of randomized and controlled clinical trials” were: (1) the assessment of biocompatibility of dental pulp-capping materials for vital pulp therapy on human dental pulp cells (in vitro) and; (2) to evaluate the inflammatory reaction and the presence of reparative dentin formation after direct pulp exposure of completely developed permanent teeth to the commercially available pulp capping materials (in vivo).
The systematic review with metanalysis is well documented and comprehensive and the conclusions are useful for clinical practice.
Author Response
Please, see attached file.

Reviewer 3 Report
The authors are to be commended for crafting such a comprehensive review. Unfortunately, the current version is considerably wordy with excessive detail and repetition. The authors should consider what additional details can be reserved for supplementary materials. There are nearly 300 lines until the reader reaches “meta-analysis” and “ objective associated results” of the literature review. Understanding the selection criteria and methods is important but details (like those is figure 1) bog down the narrative and utility of the manuscript.
There is some redundancy for example line 166 and 171 in the study selection and data collection sections. The authors are encouraged to be concise and succinct to improve the readability and impact of this very interesting manuscript.
A number of abbreviations are given (in text, tables and figures), but these are not explained. There should be a glossary of area in the text (when acronyms are first used) where these are explained.
The organization of studies in several of the tables is unclear (e.g. Table 2-4) they appear neither alphabetical (used in other figures) or chronological. As this study is an investigation of pulp-capping agents, then it is recommended that the very large tables be broken down into manageable units based on the agent(s) used.
Line 323: remove word “even” or consider revising.
Section 3.2 is overly abridged and is not informative. The results should outline major findings not exclusively refer the reader to a table.
Author Response
Please, see attached file.

Reviewer 4 Report
This manuscript examines the studies of cytotoxicity and bioactivity of pulp-capping agents towards human dental pulp cells. The inflammatory reaction and reparative dentin-bridge formation induced by different capping agents on human pulp tissue are also evaluated. It is reported that pure calcium-hydroxide powder/saline and commercial resin-free hydraulic calcium-silicate cement Pro-Root MTA are the best options that provide a reparative bridge upon vital pulp therapy. The work is impressive. The following comments should be addressed before the paper can be further considered: (a) In the first paragraph of introduction, the authors should define and briefly introduce “dental pulp-capping agents”. (b) In introduction, the sentence “The main reasons for the increase of marketed materials are the good results obtained with MTA” is not clear. What are the “good results”? (c) In introduction, the authors should also mention if any systemic review relevant to this topic has been published. (d) The authors should provide the full form of all abbreviations (e.g, MTA, MTT, ALP etc) (e) The conclusion of abstract should be improved, which is inconsistent with the conclusion of manuscript. (f) The future works should be further elaborated. Also, besides bioactivity and biocompatibility, are there any other properties to be improved?Author Response
Please, see attached file.
